# ADAM OR GAUSS-NEWTON? A COMPARATIVE STUDY IN TERMS OF BASIS ALIGNMENT AND SGD NOISE

## ABSTRACT

Approximate second-order optimizers are increasingly showing promise in accelerating training of deep learning models, yet their practical performance depends critically on how preconditioning is applied. Two predominant approaches to preconditioning are based on (1) Adam, which leverages statistics of the current gradient, and (2) Gauss-Newton (GN) methods, which use approximations to the Fisher information matrix (often raised to a power). This work compares these approaches through the lens of two key factors: the choice of basis in the preconditioner and the impact of gradient noise from mini-batching. To gain insights, we analyze these optimizers on quadratic objectives and logistic regression under all four quadrants. We show that regardless of the basis, there exist instances where Adam outperforms both $GN^{-1}$ and $GN^{-1/2}$ in full-batch settings. Conversely, in the stochastic regime, Adam behaves similarly to $GN^{-1/2}$ under a Gaussian data assumption. These theoretical results are supported by empirical studies on both convex and non-convex objectives.

## 1 INTRODUCTION

Modern deep learning has shifted away from vanilla (stochastic) gradient descent toward adaptive first-order optimizers with preconditioned updates of the form $\theta_{t+1} = \theta_t - \eta P g_t$, where the preconditioner $P \in \mathbb{R}^{d \times d}$ is often taken to be diagonal. Popular methods such as Adam (Kingma & Ba, 2014), RMSProp (Tieleman & Hinton, 2012), Adafactor (Shazeer & Stern, 2018), SignSGD (Bernstein et al., 2018), and Lion (Chen et al., 2023) all fall into this category. Recent work has shown that these optimizers perform comparably in practice (Zhao et al., 2024), prompting a natural question: what additional gains might be possible by incorporating second-order information into the diagonal preconditioner?

Second-order optimizers offer one answer to this question. Although directly using the inverse Hessian, as in Newton's method, is typically impractical at scale, a variety of scaling-friendly approximations have been proposed that are both efficient and theoretically grounded (Martens & Grosse, 2015; Gupta et al., 2018; Liu et al., 2023; Vyas et al., 2024). These methods can be viewed as first rotating the gradient into a particular basis—often derived from an approximation to the Hessian or Fisher information matrix—and then applying a diagonal preconditioner in that rotated basis. For instance, Shampoo (Gupta et al., 2018) directly approximates the Gauss-Newton algorithm—often using an exponent on the approximated eigenvalues ranging from $-1$ to $-1/4$—with a Kronecker-product structure that defines an efficient preconditioning basis (see (Morwani et al., 2024) for discussion). In contrast, SOAP (Vyas et al., 2024) can be interpreted as running Adam in a rotated basis, where the basis is potentially derived from a Kronecker-factored approximation to curvature.

This motivates a central question: can we disentangle the role of the *basis* used for preconditioning from the choice of *diagonal* scaling in that basis? In this work, we explore this question by comparing two canonical choices of diagonal scalings: one based on the running average of squared gradients, as in Adam (which approximates the diagonal of the *empirical* Fisher (Kunstner et al., 2019)), and another based on the diagonal of the Gauss-Newton (GN) matrix, which reflects curvature information derived from the Fisher or Hessian. These diagonal forms can be applied in arbitrary bases—including the identity basis (used by default in Adam) and the eigenbasis of the GN matrix. We formally define these choices in Section 2.

Table 1: Comparing Adam vs GN diagonal preconditioners across two axes: (i) basis choice, and (ii) gradient noise. Our theoretical results are based on quadratics (Section 3) and logistic regression (Section 4), across the *eigen* and *identity* bases on full (population) and small (single-sample) batch.

| | Batch-Size Regime | |
| --- | --- | --- |
| **Basis Choice** | **Full Batch** | **Small Batch** |
| Eigenbasis | $\exists$ logistic example where Adam $>$ GN$^{-1}$ | GN$^{-1} \geq$ Adam $\approx$ GN$^{-1/2}$ for quadratics |
| Identity basis | $\exists$ quadratic example where Adam $>$ GN$^{-1}$ | Adam $\approx$ GN$^{-1/2}$ for quadratics |

A guiding question in our study is whether the empirical Fisher (used in Adam and SOAP) offers any advantage over GN-derived curvature estimates, or whether it merely provides a tractable proxy. Furthermore, since Adam effectively uses the *square root* of its second-moment estimate, we also investigate whether preconditioning with GN$^{-1/2}$ is preferable to GN$^{-1}$ under various conditions. Importantly, we aim to decouple the influence of the preconditioning basis from that of the diagonal approximation applied within it.

**Our Contributions.** We study how the effectiveness of diagonal preconditioners depends on two key factors: (1) Basis choice: We compare preconditioning in the eigenbasis of the GN matrix versus in the identity basis (or more generally, bases misaligned with the eigenbasis but still satisfying structural properties such as a sparse plus low-rank structure). (2) Gradient noise: We analyze both full-batch (population gradient) and stochastic gradient (batch size 1) regimes to isolate how preconditioner behavior is influenced by gradient variance.

For linear regression, we obtain the following theoretical results:

- **Sensitivity to basis choice** (Section 3.1): It is well known that GN preconditioning in the eigenbasis yields optimal convergence rates for quadratics. However, when the basis is misaligned, Adam can outperform both GN$^{-1}$ and GN$^{-1/2}$, sometimes matching the speed of GN in its ideal basis.
- **Equivalence under noise** (Section 3.2): In the stochastic regime with Gaussian data, Adam behaves similarly to GN$^{-1/2}$ regardless of basis, suggesting a surprising alignment between its empirical design and curvature-based preconditioning.

Further, in the case of logistic regression, we show Adam can even outperform GN$^{-1}$ under the eigenbasis with full batch update (Section 4).

These results are summarized in a two-by-two grid in Table 1, and we further discuss the distinction between GN$^{-1}$ and GN$^{-1/2}$ in Section 3.3. The quadratic model and logistic regression provide complementary perspectives, yielding a more complete picture and highlighting the benefit of separating basis and gradient noise considerations.

We complement our theoretical findings with empirical results on both simulations and more general problems (Section 5). The results align with the theory across all empirical settings, illustrating the practical implications of basis choice and gradient noise. We discuss related work in Appendix A.

## 2    PRELIMINARIES

Consider optimizing a function $f : \mathbb{R}^d \to \mathbb{R}$ with input $x$ and the (vectorized) parameter $\theta \in \mathbb{R}^d$ against a loss function $\ell$. Updates performed by preconditioning optimizers can be seen as

$$\theta^{(t+1)} = \theta^{(t)} - \eta \cdot (UD^pU^\top)g^{(t)},$$

where $\eta$ denotes the learning rate, $D$ is the diagonal preconditioner which is raised to the exponent $p \in \{-\frac{1}{2}, -1\}$, $U$ is the orthonormal basis on which the preconditioned update is performed, and $g^{(t)}$ denotes the gradient at time $t$ (which could correspond to either population gradient or stochastic gradient depending on the setting). The full update is described in Algorithm 1.

---

**Algorithm 1** Preconditioned optimizer

1: **Input:** $\theta^{(0)}$, BASISTYPE $\in$ {Id, EigenBasis}, PRECOND $\in$ {Adam, GN}, power $p \in$ {$-0.5, -1$}, learning rate $\{\eta_t\}_{t=1}^T$, regularization coefficient $\epsilon$, gradient batch size $b_G$, and basis estimation batch size $b_H$.
2: **for** $t = 1$ **to** $T$ **do**
3:     Sample a batch of $b_G$ samples $X_G := \{x_i\}_{i \in [b_G]}$.
4:     Compute batch loss $\ell_t(\theta^{(t)}; X_G)$ and gradient $g^{(t)} = \nabla \ell_t(\theta^{(t)}; X_G)$.
5:     Sample a batch of $b_H$ samples $X_H := \{x_i\}_{i \in [b_H]}$.
6:     Compute the Gauss-Newton matrix $H^{(\mathrm{GN})}$ using $X_H$.
7:     **if** BASISTYPE == Id **then** // `Basis choice`
8:         Basis $U \leftarrow I$.
9:     **elif** BASISTYPE == EigenBasis **then**
10:        Basis $U \leftarrow$ EigenDecomposition($H^{(\mathrm{GN})}$).
11:     Compute the basis-rotated gradient $\tilde{g}^{(t)} = U^\top g^{(t)}$.
12:     **if** PRECOND == Adam **then** // `Diagonal preconditioner choice`
13:        Compute the diagonal preconditioner as $D_{ii} = \left(\mathbb{E}[(\tilde{g}(x)_i)^2]\right)^{-1/2}$.
14:     **elif** PRECOND == GN **then**
15:        Compute the diagonal preconditioner as $D_{ii} = (u_i^\top H^{(\mathrm{GN})} u_i)^p$.
16:     $\theta^{(t+1)} = \theta_t - UD\tilde{g}^{(t)} = \theta_t - UDU^\top g^{(t)}$.

---

We will discuss the choice of the basis and the diagonal preconditioner below. We start with describing the *Gauss-Newton* (GN) matrix, which is the first term of the Hessian:

$$H := \nabla_\theta^2 \ell = \nabla_\theta f \nabla_f^2 \ell \nabla_\theta f^\top + \nabla_f \ell \nabla_\theta^2 f := H^{(\mathrm{GN})} + \nabla_f \ell \nabla_\theta^2 f, \qquad (1)$$

Contrast to the Hessian $H$, the GN matrix $H^{(\mathrm{GN})}$ requires only first-order gradient information to compute and often serves as a reasonable preconditioner in practice (Sankar et al., 2021). For convex loss functions, which will be the focus of this work, the GN term is positive-semidefinite (PSD) and admits a real-valued eigendecomposition.

**Basis estimation.** Our theory analyzes two basis choices: 1) the identity basis $U = I$, and 2) the eigen-basis of the Gauss-Newton matrix $H^{(\mathrm{GN})}$; for the mean-squared loss, $H^{(\mathrm{GN})} = \mathbb{E}_x[\nabla_\theta f(x) \nabla_\theta f(x)^\top]$. For the experiments, we additionally consider the Kronecker-factored preconditioner (Martens & Grosse, 2015; Vyas et al., 2024) as a computationally efficient approximation of the eigenbasis: for a matrix-valued parameter $\theta \in \mathbb{R}^{n \times m}$, where $H^{(\mathrm{GN})} \in \mathbb{R}^{nm \times nm}$ is approximated by the outer product of two matrices of dimension $\mathbb{R}^{n \times n}$ and $\mathbb{R}^{m \times m}$; see Appendix D.1 for details.

**Diagonal preconditioners.** Given an orthonormal basis $U$, we first rotate the gradient $g$ into the basis $\tilde{g} := U^\top g$, then apply a diagonal conditioner $D$ to the rotated gradient. We consider two types of $D$: Adam and Gauss-Newton. For the theoretical part, we are going to consider the Adam's preconditioner as described below:

$$D_{ii}^{(A)} := \left(\mathbb{E}[(\tilde{g}(x)_i)^2]\right)^{-1/2} = \left(\mathbb{E}[(u_i^\top g(x))^2]\right)^{-1/2}, \qquad (2)$$

where $g(x)$ denotes the gradient from $x$, and the expectation is over all possible batches. Note that in the full batch case, this simply corresponds to the rotated gradient, while in the stochastic case, it represents the expected per-sample gradient norm. This is motivated from the practical version of Adam, which maintains a running average of the gradients seen during the training. For Gauss-Newton, it takes an additional exponent parameter $p \in \{-\frac{1}{2}, -1\}$ and computes the diagonal elements as

$$D_{ii}^{(\mathrm{GN})} := (u_i^\top H^{(\mathrm{GN})} u_i)^p, \qquad (3)$$

where $u_i$ is the $i_{th}$ vector in the given basis. In particular, when $U$ is the eigenbasis of $H^{(\mathrm{GN})}$, $\{u_i^\top H^{(\mathrm{GN})} u_i\}_{i \in [d]}$ give the eigenvalues of $H^{(\mathrm{GN})}$.

The preconditioners for Adam an Gauss-Newton correspond respectively to *empirical Fisher matrix* and the *Fisher matrix*. The former is defined with gradients with respect to labels from the true data distribution, whereas the latter is defined with respect to the output of the model.

## 3 Theoretical Analysis on Linear Regression

This section focuses on linear regression with the mean-squared loss, where we compare Adam and Gauss-Newton along both axes in Table 1: the quality of the basis choice, and the effect of batch size (gradient noise). The goal is to learn a function $f_\theta(x) = \theta^\top x$ with loss $\ell(\theta) = \frac{1}{2}\mathbb{E}[((\theta - \theta^*)^\top x)^2]$, where $\theta^*$ is the ground truth parameter. We consider Gaussian input $x \sim \mathcal{N}(0, \Sigma_x)$. For this setup, the vanilla gradient descent update has $\Delta^{(t+1)} := \theta^{(t+1)} - \theta^* = (\mathbf{I} - \eta\Sigma_x)\Delta^{(t)}$, and the Gauss-Newton matrix simplifies to the data covariance matrix, i.e. $H^{(\text{GN})} = \mathbb{E}[\nabla_\theta f \nabla_\theta f^\top] = \Sigma_x$.

### 3.1 Axis 1 – Choice of basis

Deep learning optimization often encounters heterogeneous curvature, which motivates the use of preconditioned methods (Sagun et al., 2016; Ghorbani et al., 2019; Yao et al., 2019; Zhang et al., 2020; Liu et al., 2023). However, while $\text{GN}^{-1}$ is known to achieve optimal convergence under the correct eigenbasis (Section 3.1.1), we show in Section 3.1.2 that $\text{GN}^{-1}$ can be severely suboptimal under a poorly chosen basis that does not reflect the true curvature. In contrast, Adam has an "auto-tuning" effect that allows it to outperform both $\text{GN}^{-1}$ and $\text{GN}^{-\frac{1}{2}}$. We support this with a constructed example.

#### 3.1.1 $\text{GN}^{-1}$ is optimal under the correct basis

The basis given by the eigenvectors of $H^{(\text{GN})}$ can be considered as the "correct" basis. Under this basis, it is well-known that $\text{GN}^{-1}$ achieves the optimal convergence rate in both full-batch and stochastic setting [1] : when using the full batch, $\text{GN}^{-1}$ converges in 1 step; for the stochastic setting, $\text{GN}^{-1}$ decreases the loss at a linear rate. Details are included in Appendix B.1 for completeness.

#### 3.1.2 Incorrect basis: Adam auto-tunes to the curvature

The previous section shows that GN is optimal under the ideal eigenbasis. What if the basis is not estimated correctly? In this section, we show that GN is sensitive to the basis choice. We provide an example where the wrong basis obscures the true curvature, voiding GN's adaptiveness.

Consider a quadratic problem with a covariance

$$\Sigma_x := \mathbb{E}[xx^\top] = \begin{bmatrix} \mathbf{1}\mathbf{1}^\top & \mathbf{0} \\ \mathbf{0} & \mathbf{I} \end{bmatrix} \in \mathbb{R}^{2d \times 2d}, \tag{4}$$

where $\mathbf{1} \in \mathbb{R}^d$ is the all-one vector, and $\mathbf{I} \in \mathbb{R}^{d \times d}$ is the identity matrix. This problem has a block structure that is symmetric among the first $d$ coordinates and among the last $d$ coordinates. The two blocks have widely different maximal eigenvalues and hence different optimal learning rates: The first block has a maximum eigenvalue of $d$, and thus the maximum stable learning rate is the $\frac{2}{d}$. In contrast, all eigenvalues for the second block are 1, hence the maximum learning rate affordable is 2.

The wrong basis choice we consider is the identity basis. We will see that this makes GN fail to adapt to the curvature of the problem, while Adam remains efficient.

**GN converges slowly.** When taking $U = I$, the diagonal preconditioner for GN has $D_{ii}^{(\text{GN})} = 1^p = 1$. That is, both $\text{GN}^{-1}$ and $\text{GN}^{-\frac{1}{2}}$ simply scale all coordinates by the same factor as all diagonal elements are 1, behaving the same as vanilla gradient descent.

**Adam "auto-tunes" to the curvature.** Given the symmetry of the problem, Adam effectively acts as normalized gradient descent for the first block, and acts as signed gradient descent in each coordinate in the second block. As we train using Adam with a constant learning rate $\eta$, let $\|g_0^{(t)}\|$ represent the gradient norm for the first block coordinates, and $|g_i^{(t)}|$ for $i \in [d]$ represent the per-coordinate gradient norm for the coordinates in the second block which evolves independently.

---

[1]By stochastic setting we refer to updates where each batch contains a single sample.

Recall that for quadratic problem, the gradient is $g = \Sigma_x \Delta$ where $\Delta := \theta - \theta^*$, and $\Delta^{(t+1)} = (\mathbf{I} - \eta \Sigma_x)\Delta^{(t)}$ for vanilla gradient descent; the gradient norm goes down as long as $\eta \le \frac{2}{\lambda_{\max}}$. For Adam, the updates can be considered as gradient descent with an adaptive learning rate. Specifically, following Equation (2), the first block updates as $\Delta_0^{(t+1)} = (\mathbf{I} - \frac{\eta}{\|g_0^{(t)}\|_2} \cdot \Sigma_x)\Delta_0^{(t)}$, and the second block has per-coordinate updates $\Delta_i^{(t+1)} = (\mathbf{I} - \frac{\eta}{|g_i^{(t)}|_2} \cdot \Sigma_x)\Delta_i^{(t)}$ for $i \in [d]$. This means:

1. $\|g_0^{(t)}\|$ decreases provided $\eta/\|g_0^{(t)}\| \le 2/d$,

2. $|g_i^{(t)}|$ decreases for $i > 0$ provided $\eta/|g_i^{(t)}| \le 2$.

Thus essentially, after an initial "burn-in" period, $\eta/\|g_0^{(t)}\|$ reaches $2/d$ and oscillates around this value, while $\eta/|g_i^{(t)}|$ oscillates around 2. We refer to this as the *auto-tuning* of Adam, where it adapts to the curvature of different coordinates on its own, by regulating the gradient norms. After this burn-in period, we can reduce the learning rate by half at every step, and reach a target error within log number of steps. We note that this *auto-tuning* effect is similar to adapting to the smoothness of the curvature, which is known to be a property of normalized gradient descent (Orabona, 2023). Note that auto-tuning comes from the norm of the mean gradient in the Adam's denominator for the full batch case. Instead, Adam behaves similarly to $\mathrm{GN}^{-1/2}$ at small batch sizes, as we will show in Section 3.2, and does not exhibit this autotuning effect when the gradient variance dominates. [2] Concurrent work by Roulet & Agarwala (2025) provides consistent empirical evidence that at small batch sizes, keeping only the mean term in the Adam's denominator improves its performance, thus supporting the auto-tuning viewpoint of Adam.

### 3.2   AXIS 2 – GRADIENT NOISE AND BATCH SIZE

The previous section shows an example where Adam outperforms GN when the updates are using population gradients but a poor basis choice. In this section, we focus on the stochastic regime and show that Adam and $\mathrm{GN}^{-\frac{1}{2}}$ behave similarly regardless of the basis choice. Proofs for this section can be found in Appendix B.2.

We first prove a stronger result, showing that for Gaussian input distribution and quadratic loss, empirical Fisher is approximately equal to Fisher up to a loss scaling.

**Lemma 1.** *For linear regression with Gaussian inputs, the following holds:*

$$\ell(\theta) \cdot \Sigma_x \preceq \frac{1}{2}\mathbb{E}[g(x)g(x)^\top] \preceq 3\ell(\theta) \cdot \Sigma_x.$$

We can utilize the lemma above to show that $\mathrm{GN}^{-\frac{1}{2}}$ and Adam behave the same upto a scalar constant in any basis for our theoretical setup at batch size 1.

**Corollary 1.** *For single-sample updates, the update of Adam and $\mathrm{GN}^{-\frac{1}{2}}$ differ by a constant.*

$$\frac{1}{\sqrt{3\ell}} \cdot D^{(GN, -\frac{1}{2})} \preceq \frac{1}{2}D^{(A)} \preceq \frac{1}{\sqrt{\ell}} \cdot D^{(GN, -\frac{1}{2})}.$$

From the above lemmas, we expect Adam and $\mathrm{GN}^{-1/2}$ to have a similar performance for small batch size. However, we still don't know if $\mathrm{GN}^{-1}$ performs better than $\mathrm{GN}^{-1/2}$ at small batch. To answer this, we provide a lemma quantifying the convergence rate of a general preconditioner $P$ for linear regression with stochastic Gaussian inputs. Denoting the preconditioned Hessian as $\boldsymbol{A}(P) := P^{1/2}\Sigma_x P^{1/2}$, we have that:

**Lemma 2.** *For a general preconditioner $P$, for linear regression with stochastic Gaussian inputs, the following holds:*

$$\mathbb{E}[\ell^{(t)}] \le O\left[\left(1 - \frac{\lambda_{\min}(\boldsymbol{A}(P))}{3\operatorname{Tr}(\boldsymbol{A}(P))}\right)^t \ell^{(0)}\right].$$

---

[2]Recall the definition of $D_{ii}^{(A)}$ from Equation (2). For the $i_{th}$ basis vector $u_i$, let $\mu_i := u_i^\top \mathbb{E}_x[g(x)]$ and $\sigma_i^2 := \mathbb{E}[(u_i^\top g(x) - \mu_i)^2]$ denote the mean and variance of the gradient projection along $u_i$. Consider gradients computed on a batch of size $B$, then $(D_{ii}^{(A)})^2 = \mu_i^2 + \frac{1}{B}\sigma_i^2$. $(D_{ii}^{(A)})^2$ is dominated by the mean for full-batch updates ($B \to \infty$), and is often dominated by the variance in the stochastic regime ($B = 1$).

Defining a version of the condition number $\kappa_s(\boldsymbol{A}(P)) = \frac{\text{Tr}(\boldsymbol{A}(P))}{\lambda_{\min}(\boldsymbol{A}(P))} = \sum_{i \in d} \frac{\lambda_i(\boldsymbol{A}(P))}{\lambda_{\min}(\boldsymbol{A}(P))} \geq d$. We can see that the convergence rate depends on $\kappa_s(\boldsymbol{A}(P))$.

The above bound shows that in the correct basis, $\text{GN}^{-1}$ is the optimal preconditioner even in the stochastic regime, as it minimizes the condition number to $\kappa_s(\boldsymbol{A}(P)) = \kappa_s(\mathbf{I}) = d$. For $\text{GN}^{-1/2}$ in the correct basis, we have $\kappa_s(\boldsymbol{A}(P)) = \kappa_s((\Sigma_x)^{1/2}) = \sum_i \sqrt{\frac{\lambda_i(\Sigma_x)}{\lambda_{\min}(\Sigma_x)}}$, which is greater than $d$ unless $\lambda_{\max}(\Sigma_x) = \lambda_{\min}(\Sigma_x)$.

### 3.3 WHICH POWER TO USE FOR GAUSS-NEWTON?

Newton's method was originally proposed with a preconditioner closer to $\text{GN}^{-1}$. However, the square root in the denominator of Adam (Kingma & Ba, 2014) and Adagrad (Duchi et al., 2011) has spurred multiple papers (Liu et al., 2023; Lin et al., 2024; Vyas et al., 2024) questioning the correct power of the preconditioner to be used in practice. As shown before, for the quadratic model, in the correct basis, $\text{GN}^{-1}$ is optimal. However, under the incorrect identity basis, we claim that there exists problems for which, even in the full batch case, $p = 0.5$ leads to faster convergence than $p = 1$. The key idea is that the convergence rate of preconditioned gradient descent (Boyd & Vandenberghe, 2004) depends on the condition number of the preconditioned Hessian. It then suffices to construct examples where the condition number is better behaved for $p = 0.5$ than $p = 1$. We provide details in Appendix B.3 and accompanying simulation results in Section 5.

## 4 THEORETICAL ANALYSIS ON LOGISTIC REGRESSION

For quadratics, Section 3.1.1 shows that $\text{GN}^{-1}$ is optimal under the eigenbasis. However, this needs not hold in general. In this section, we provide an example with logistic regression, where Adam converges faster than $\text{GN}^{-1}$ even under the eigenbasis with full batches.

**Setup.** Since we are operating under the eigenbasis, the updates can be considered per dimension. We hence take the input $x$ from the set of $d$-dimensional one-hot vectors $\{e_i\}_{i=1}^d$, with probability $\nu_i := \mathbf{Pr}(x = e_i)$. Conditional on $x = e_i$, the label $y$ is Bernoulli with mean $P_i := \mathbf{Pr}(y = 1 | x = e_i)$. We assume $0.6 \leq P_i \leq 0.8$, $\forall i \in [d]$, i.e. the labels are neither deterministic nor fully random, and the optimal parameter has a bounded norm.

We optimize a *two-layer linear network* $q : \mathbb{R}^d \to \mathbb{R}^d$, whose output depends on the *squares* of the weights: for any $\theta \in \mathbb{R}^d$ and for $i \in [d]$, we define the model's prediction as

$$q_i(\theta) = \mathbf{Pr}_\theta(y = 1 \mid x = e_i) = \sigma\Big(\sum_{j=1}^d \theta_j^2 x_j\Big) = \sigma(\theta_i^2), \qquad \sigma(z) = \frac{1}{1 + \exp(-z)}. \tag{5}$$

The square parameterization makes the problem non-convex, and is analogous to the structure of key-query multiplication in self-attention (Vaswani et al., 2017).

In the following, we show that under the natural assumption of non-increasing step sizes, there is a separation between Adam and $\text{GN}^{-1}$ in terms of $\kappa(\nu) := \frac{\nu_{\max}}{\nu_{\min}}$. We consider local convergence near the optimum. In particular, let $\kappa(\nu) = \Omega(d^{1/2+\delta})$ for some $\delta \in [0, \frac{1}{2}]$, and let $\epsilon$ denote the target parameter error, i.e. we want to find $\theta$ such that $\|\theta - \theta^*\|_2 \leq \epsilon$. We prove that Adam enjoys dimension-free convergence, whereas $\text{GN}^{-1}$ suffers from a polynomial-in-dimension slowdown.

**Adam converges in $O(\log(1/\epsilon))$ steps.** Adam effectively performs sign GD, hence the amount of parameter update is determined by the step size. The $O(\log(1/\epsilon))$ convergence hence follows directly from starting with a $O(1)$ learning rate and halving the step size every $O(1)$ steps.

**$\text{GN}^{-1}$ requires $\tilde{\Omega}(d^\delta \log(1/\epsilon))$ steps.** In contrast to Adam, GN's precondtioning can result in an unboundedly large update (see Equation (21)), unless the learning rate is kept small, which in turn leads to slow convergence. To show the lower bound, we first state a more general convergence result. Recall that the $\text{GN}^{-1}$ update is $\theta^{(t+1)} = \theta^{(t)} - \eta^{(t)}(H^{(\text{GN})}(\theta) + \alpha I)^{-1} g^{(t)}$. We assume that

$\{\eta^{(t)}\}_{t\geq0}$ are non-increasing and that the step size schedule and regularization lead to convergence, i.e. $\theta^{(t)} \to \theta^*$ as $t \to \infty$. Linearizing the update map at the limit $\theta^*$ gives

$$\theta^{(t+1)} - \theta^* = \Big(I - \eta^\infty (H_*^{(\text{GN})} + \alpha I)^{-1} H_*^{(\text{GN})}\Big)(\theta^{(t)} - \theta^*) \;+\; O\big(\|\theta^{(t)} - \theta^*\|^2\big),$$

where $H_*^{(\text{GN})} := H^{(\text{GN})}(\theta^*)$ and $\eta^\infty = \lim_{t\to\infty} \eta_t$. We are interested in lower bounding the spectral radius of the local iteration matrix:

$$\gamma(\eta^{(\infty)}, \alpha) := \big\| I - \eta^{(\infty)}(H_*^{(\text{GN})} + \alpha I)^{-1} H_*^{(\text{GN})}\big\|_2,$$

as it governs the ultimate *local* rate of convergence that any GN schedule can achieve. We refer to $\gamma(\eta^{(\infty)}, \alpha)$ as the local contraction factor; a value of $\gamma$ close to 1 implies slow convergence.

The main result of this section is a lower bound on $\gamma$:

**Theorem 2.** *Suppose the weights are initialized at $\theta^{(0)} = \frac{1}{\sqrt{d}} \cdot \vec{1}$. Consider any non-increasing step size sequence $\{\eta^{(t)}\}_{t\geq0}$ and regularization parameter $\alpha \geq 0$. If the Gauss-Newton iterates converge to $\theta^*$, i.e. $\theta^{(t)} \to \theta^*$, then the local contraction factor $\gamma(\eta^{(\infty)}, \alpha)$ is lower bounded by:*

$$\gamma(\eta^{(\infty)}, \alpha) \geq 1 - c\sqrt{\log d} \, \max\left\{ \frac{1}{\sqrt{d}}, \sqrt{d/\kappa(\nu)} \right\},$$

*for some universal constant $c$.*

This theorem reveals a basic trade-off: for the Gauss-Newton method to converge globally from our chosen starting point, its final learning rate must be small. This restriction, in turn, hampers its local convergence speed and creates a bottleneck. The slowdown is substantial under the common conditions of high dimensionality and ill-conditioned data:

**Corollary 3.** *For imbalanced input with $\kappa(\nu) = \Omega(d^{1/2+\delta})$ for some $\delta \in [0, 1/2]$, $GN^{-1}$ requires $t = \tilde{\Omega}(d^\delta \log(1/\epsilon))$ steps to reach a parameter satisfying $\|\theta^{(t)} - \theta^*\|_2 \leq \epsilon$.*

This demonstrates a polynomial slowdown in the dimension, highlighting a scenario where the theoretical power of Gauss-Newton is significantly degraded due to the practical requirement of global convergence. We empirically verify this on Transformers in Section 5.

*Remark*: Ideally, in a purely local setting, one could choose $\alpha = 0$ and use a constant final stepsize $\eta^\infty$. For instance, setting $\eta^\infty = 1$ would make the iteration matrix zero, yielding $\gamma = 0$ and superlinear convergence. In fact, any other constant $\eta^\infty \in (0, 2)$ would similarly provide rapid linear convergence with a rate independent of the condition number. However, Theorem 2 shows this ideal scenario is not possible. As the proof (Appendix C) shows, the requirement of ensuring convergence from a specific, natural initialization forces the algorithm's final step size, $\eta^{(\infty)}$, to be small. This constraint directly degrades the local contraction factor, preventing the rapid convergence one might expect from a Newton-like method. Further, we note that our result does not contradict with the fast convergence from line search, which does not fall under the non-increasing step size assumption.

## 5 EXPERIMENTS

In this section, we provide experimental evidence to the theoretical claims made in the previous section. Our experiments are broadly divided into two categories: (i) simulations for examples in Section 3 and Section 4, (ii) non-convex examples with MLP, and (iii) Transformer experiments.

**Experiment details.** We use the mean square error as the objective function unless otherwise specified. For numerical stability, we optionally regularize $D$ to be $D + \alpha I$ for some small $\alpha > 0$. We sweep over the learning rate $\eta$ and the regularization coefficient $\alpha$. For Adam, we also sweep over the learning rate schedule (constant or step decay) and $\beta_2$; we fix $\beta_1 = 0$, similar to Das et al. (2024). We use different samples for estimating gradient and Gauss-Newton. Due to computational considerations, our eigenbasis experiments also consider Kronecker approximation of the full eigenbasis. We report the mean and standard error based on 10 seeds. More details are provided in Appendix D.1.

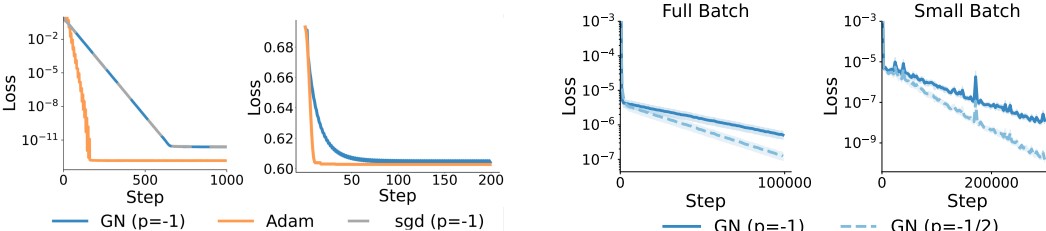

Figure 1: **Simulation results**. (*Left*) Adam converges faster than GN with full batches 1) under the identity basis on a linear regression task with block-wise covariance (left subplot), where GN fails to adapt to the problem curvature and behaves identically to SGD; and 2) under the eigenbasis, for the reparameterized logistic regression task (right subplot). (*Right*) Comparing GN power $p \in \{-\frac{1}{2}, -1\}$. On a regression task where $\text{GN}^{-\frac{1}{2}}$ leads to a more favorable condition number, $\text{GN}^{-\frac{1}{2}}$ converges faster than $\text{GN}^{-1}$ with both small and large batches.

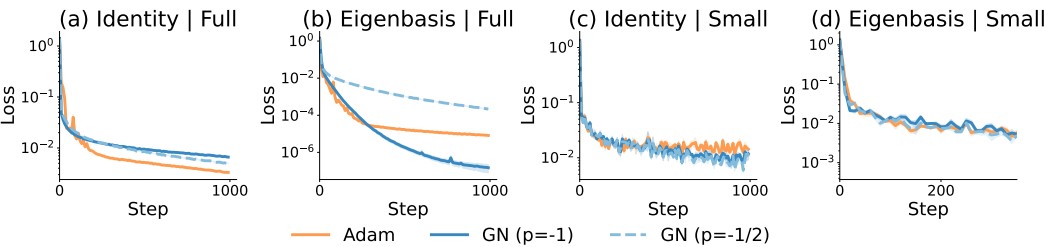

Figure 2: **Learning from a random teacher**, comparing Adam, $\text{GN}^{-1}$ and $\text{GN}^{-1/2}$ for the full $2 \times 2$ grid (Table 1).

**Simulations**    This section provides simulation results on the examples in Section 3 and Section 4.

- *Comparing Adam and GN under full-batch updates.* We empirically verify the examples provided in the theory where Adam can outperform GN both under the identity basis and the eigenbasis (Figure 1 left). For the identity basis, we consider a 100-dimensional linear regression task (Section 3.1.2) where the covariance matrix has a block-wise structure. Adam converges quickly due to the auto-tuning effect, whereas Gauss-Newton converge as slowly as vanilla gradient descent. For logistic regression (Section 4), our results on a 2048-dimensional problem (detailed in Appendix D.4) show that Adam converges faster even under the eigenbasis.

- *Comparing powers of Gauss-Newton.* Section 3.3 shows that the comparison of $\text{GN}^{-1}$ and $\text{GN}^{-\frac{1}{2}}$ amounts to comparing the condition number of a particular matrix, for both population and stochastic settings. We empirically verify the claim on a 5-dimensional linear regression problem, controlling the choice of the covariance $\Sigma_x$. Figure 1 (right) shows the simulation results in this case, where $\text{GN}^{-\frac{1}{2}}$ converges faster than $\text{GN}^{-1}$ when using both large and small batches. Details are provided in Appendix B.3.

**Non-convex examples with MLP**    Next, we consider non-convex optimization with one-hidden-layer MLPs on the following tasks: 1) learning from a random teacher network; 2) feature learning with sparse parity and its variant "staircase", where the labels depend on a subset of input coordinates; and 3) CIFAR10 image classification, where the class labels are treated as one-hot vectors; Appendix D.2 provides details. Experiments on these tasks cover the full $2 \times 2$ grid in Table 1, with respect to batch size (full vs small) and the basis (eigenbasis vs identity). We use the Kronecker approximation for the eigenbasis of the Hessian, which behaves similarly as the full eigenbasis (Appendix D.1) while being more compute efficient. We provide additional experiments on intermediate basis choices by interpolating between the identity and the eigenbasis in Appendix D.3.

As shown in Figure2–4, our theoretical analyses empirically extend to these four non-convex tasks across both axes of interest. In particular, across all problems considered, Adam and $\text{GN}^{-1/2}$ closely track one another at small batch sizes, regardless of the basis chosen (subfigures (c), (d)). Moreover, Adam is close to or better than $\text{GN}^{-1}$ when the basis is incorrect (subfigures (a), (c)).

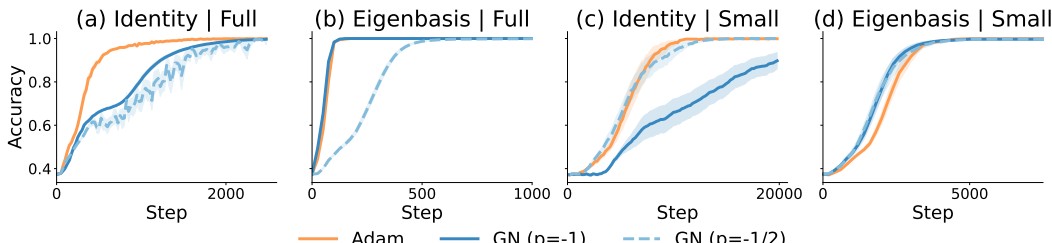

Figure 3: **Staircase**, comparing Adam, $GN^{-1}$ and $GN^{-1/2}$ for the full $2 \times 2$ grid (Table 1). Staircase is a generalization of sparse parity (Figure 7).

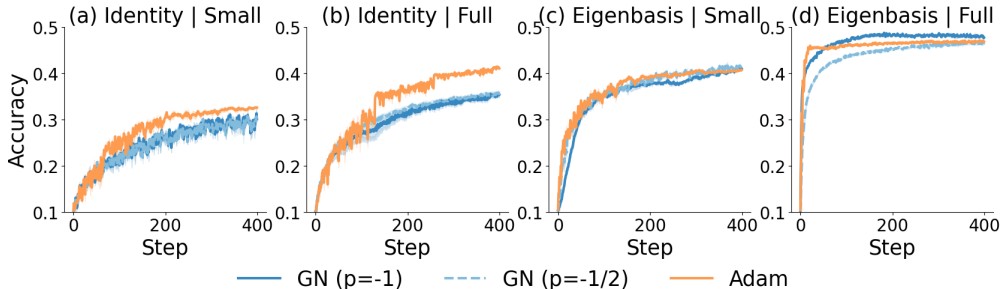

Figure 4: **CIFAR10**, comparing Adam, $GN^{-1}$ and $GN^{-1/2}$ for the full $2 \times 2$ grid (Table 1).

**A logistic-like task with Transformers** We consider a selection-based regression task learned with a 1-layer Transformer, whose attention module resembles the structure of the reparametrized logistic regression in Section 4 (detailed in Appendix D.4). Results in Figure 5 are consistent with our theory, where Adam outperforms GN under the eigenbasis with full batch updates.

## 6 DISCUSSION

This work studies the effectiveness of diagonal preconditioners along two key factors: the alignment to the ideal eigenbasis, and the level of gradient noise as influenced by the batch size. Our theoretical results on linear and logistic regression show that the comparison between Adam and Gauss-Newton (GN)-based diagonal preconditioners is sensitive to the change in either factor (Table 1): In the full batch setting, Adam can outperform GN in the identity basis for linear regression, and can even outperform GN in the ideal eigenbasis when considering logistic regression. In contrast, in the stochastic regime, we show that Adam and $GN^{-\frac{1}{2}}$ exhibit similar behavior for linear regression regardless of the basis choice, thereby revealing a connection between Adam's design and curvature-based preconditioning.

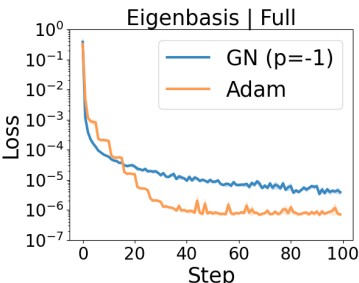

Figure 5: Transformer experiments (Section 5): Adam outperforms $GN^{-1}$ under the eigenbasis with full batches.

It is important to understand whether phenomena and differences observed in small-scale, synthetic setups persist across scale. Our empirical results on simulation, synthetic, and image datasets are consistent with the theoretical results. In particular, all MLP experiments align with findings for linear regression, and the Transformer experiments align with our logistic regression results. A more thorough study of implications on larger-scale neural networks is an interesting next step. For instance, it is been observed that layer-based block-wise approximation of the full Gauss-Newton matrix can lead to similar or sometimes superior performance Benzing (2022); Zhang et al. (2025); Abreu et al. (2025).

Moreover, we hypothesize that the equivalence between Adam and $GN^{-\frac{1}{2}}$ in the stochastic regime extends to practical, large-scale training. In particular, as training progresses, the gradient variance

tends to dominate over the gradient mean, mirroring the stochastic regime in which variance drives the dynamics. Validating this hypothesis in large-scale settings is an interesting direction for future work. Finally, such equivalence combined with the benefits of Adam's *auto-tuning* at large-batch regimes suggests a promising direction: developing algorithms that exhibit similar desirable auto-tuning behavior even when operating with small batches.

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

**The Use of Large Language Models**   In this work, LLMs are used for rephrasing and polishing the writing, assisting with basic algebraic manipulation in the proof, and assisting with the plotting code.

## A   RELATED WORK

**Approximate second-order optimizers**   Recent advances on approximate second-order methods have demonstrated success in large-scale settings, serving as efficient alternatives to classic second-order methods such as Newton's and natural gradient descent, which are computationally bottle-necked to scale to high dimensions. While first-order diagonal preconditioners are shown to be comparable in practice Kaddour et al. (2023); Zhao et al. (2024), leveraging second-order information has proven effective. Most relevant to our work are methods that can be considered as applying diagonal preconditioners in a chosen basis (Gupta et al., 2018; Liu et al., 2023; Vyas et al., 2024; Jordan et al., 2024). However, there is no clear understanding how the preconditioner and the basis interact. For instance, Sophia (Liu et al., 2023) can be considered as applying $\mathrm{GN}^{-1}$ in the identity basis, which as we will show, is not always desired. In contrast, our work provides a clarifying decomposition of the design space of these second-order methods by separating the choice of basis in which to perform a diagonal preconditioner, from the choice of the diagonal preconditioner itself.

For results in the stochastic regime, Martens (2014) provided results the stochastic case and depends on the condition number, similar to Lemma 2. However, their setting crucially differs from ours by assuming that the covariance of the gradients is independent of the current iterate.

**Efficient optimizers for large-scale training**   Although preconditioned methods are theoretically appealing for their faster convergence, substantial efforts have been made to translate these gains to practical speedups in wall-clock time, which is crucial in modern large-scale training. To keep each update step lightweight, it is common to approximate the Hessian using the Gauss-Newton matrix (Equation (1)), which, despite being biased, relies only on gradient information and is therefore more computationally efficient than the other commonly used Hutchinson estimator. In addition, when estimating the eigenbasis, one can use the Kronecker factorization in place of the full basis (Martens & Grosse, 2015; George et al., 2018; Vyas et al., 2024). In this work, we analyze the full eigenbasis of the Gauss-Newton matrix, while adopting the Kronecker approximation in the experiments (see Appendix D.1 for details).

**Adam vs (S)GD**   There have been a lot of interest understanding the comparison of Adam and (stochastic) gradient descent. Related to the preconditioning perspective in our work, Das et al. (2024) studies Adam's preconditioning effect on quadratics, and shows that it outperforms SGD when the Hessian is sufficiently ill-conditioned. A line work focuses the comparison on optimizing Transformers, which has investigated through the lens of gradient noises (Zhang et al., 2020), relation to sign descent (Kunstner et al., 2023), and the curvature of the landscape (Jiang et al., 2023; Pan & Li, 2023); Ahn et al. (2024) provides a review and a theory-friendly abstraction. Most related to our work is Maes et al. (2024), which shows that Adam's advantage over SGD for Transformers rely crucially on the choice of basis. While these results hinge on properties specific to Transformers, we are interested in understanding of algorithm design with insights that can be generally applicable.

## B   THEORETICAL RESULTS AND OMITTED PROOFS

### B.1   OPTIMALITY OF $\mathrm{GN}^{-1}$ IN THE CORRECT BASIS

For completeness, we provide proofs for the optimality of $\mathrm{GN}^{-1}$ for the quadratic loss under the correct eigenbasis.

**Full batch**   For $\mathrm{GN}^{-1}$, the parameter estimation error evolves as

$$\begin{aligned}
\theta^{(1)} - \theta^* &= (\theta^{(0)} - \theta^*) - \eta \cdot (H^{(\mathrm{GN})})^{-1} \cdot g^{(0)} \\
&= \theta^{(0)} - \eta \cdot \mathbb{E}[xx^\top]^{-1} \cdot \mathbb{E}[xx^\top(\theta - \theta^*)] = (1 - \eta)(\theta - \theta^*).
\end{aligned} \tag{6}$$

Hence $\mathrm{GN}^{-1}$ can reach the optimum in 1 step with $\eta = 1$.

**Stochastic regime**  Let's consider the stochastic regime where each update step is performed with a single sample. With respect to some algorithm, define

$$M^{(t)} := \mathbb{E}[(\theta^{(t)} - \theta^*)(\theta^{(t)} - \theta^*)^\top].$$

which is the expected second-moment matrix of the distance-to-opt.

Recall that $\text{GN}^{-1}$ has updates $\theta^{(t+1)} = \theta^{(t)} - \eta \Sigma_x^{-1} g^{(t)}$, where $g = (\theta - \theta^*)^\top x \cdot x$. We have that:

$$M^{(t+1)} = M^{(t)} - 2\eta M^{(t)} + \eta^2 \text{Tr}(M^{(t)} \Sigma_x) \Sigma_x^{-1} + 2\eta^2 M^{(t)}. \tag{7}$$

Multiplying by $\Sigma_x$ and taking the trace leads to:

$$\mathbb{E}[\ell^{(t+1)}] = \big(1 - 2\eta + 2\eta^2(d+1)\big) \mathbb{E}[\ell^{(t)}]. \tag{8}$$

Setting $\eta = \frac{1}{2(d+1)}$ reduces the expected error by a $\frac{1}{2}$ factor every $O(d)$ steps.

## B.2  EQUIVALENCE OF ADAM AND $\text{GN}^{-0.5}$ UNDER STOCHASTIC REGIME

We start with establishing the equivalence between the empirical and true Fisher (Lemma 1), which will then be used to prove the equivalence of Adam and $\text{GN}^{-\frac{1}{2}}$'s updates.

### B.2.1  PROOF OF LEMMA 1: EQUIVALENCE OF THE EMPIRICAL AND TRUE FISHER

**Lemma** (Lemma 1, restated). *For linear regression with Gaussian inputs, the following holds:*

$$\ell(\theta) \cdot \Sigma_x \preceq \frac{1}{2} \mathbb{E}[g(x)g(x)^\top] \preceq 3\ell(\theta) \cdot \Sigma_x.$$

*Proof.* For a given $\theta$, let $M := (\theta - \theta^*)(\theta - \theta^*)^\top$. Then w.r.t. $\theta$, we have

$$\mathbb{E}[g(x)g(x)^\top] = \mathbb{E}[x^\top M x \cdot xx^\top] = 2\Sigma_x M \Sigma_x + \text{Tr}(\Sigma_x M)\Sigma_x \preceq 3\text{Tr}(\Sigma_x M)\Sigma_x, \tag{9}$$

where the last equality follows from Wick's theorem. The lemma follows by noting that $\ell = \frac{1}{2}\mathbb{E}[x^\top M x] = \frac{1}{2}\text{Tr}(\Sigma_x M)$. $\qquad\square$

### B.2.2  PROOF OF COROLLARY 1: EQUIVALENCE OF ADAM AND $\text{GN}^{-\frac{1}{2}}$

**Corollary** (Corollary 1, restated). *For single-sample updates, the update of Adam and $\text{GN}^{-\frac{1}{2}}$ differ by a constant.*

$$\frac{1}{\sqrt{3\ell}} \cdot D^{(GN, -\frac{1}{2})} \preceq \frac{1}{2} D^{(A)} \preceq \frac{1}{\sqrt{\ell}} \cdot D^{(GN, -\frac{1}{2})}.$$

*Proof.* Adam's preconditioner is based on

$$P^{(A)} := \mathbb{E}_{(x,y) \sim \mathcal{D}}[g(x)g(x)^\top]. \tag{10}$$

Given a basis $U$, the diagonal preconditioner given by Adam has entries

$$(D_{ii}^{(A)})^{-1} = \sqrt{u_i^\top P^{(A)} u_i}. \tag{11}$$

The relation between $D^{(A)}$ and $D^{(\text{GN}, -0.5)}$ follows from Lemma 1 and the fact that $(D_{ii}^{(\text{GN}, -0.5)})^{-1} = \sqrt{u_i^\top \Sigma_x u_i}$.

$$\square$$

**Part 2: when $H^{(\mathbf{GN})}$ is based on a single sample** On single-sample batches, the gradient is $g(\theta) = \ell_f' \cdot \nabla_\theta f$. Then, the diagonal preconditioner for Adam (with $\beta_1 = \beta_2 = 0$) has Given a basis $U$, let $\tilde{g} := U^\top g$ denote the gradient rotated into the basis.

$$D_{ii}^{(A)} = (|\tilde{g}_i|)^{-1} = \left((\ell_f')^2 \cdot (U^\top \nabla_\theta f_i)^2\right)^{-0.5}.$$

The diagonal preconditioner for Gauss-Newton has

$$D_{ii}^{(GN)} = (H_{ii}^{(\mathbf{GN})})^{-0.5} = (\ell_f'' \cdot (U^\top \nabla_\theta f_i)^2)^{-0.5} = ((\ell_f')^2 / \ell_f'')^{0.5} \cdot D_{ii}^{(A)}.$$

### B.2.3 PROOF OF LEMMA 2: LOSS CONVERGENCE OF GENERAL PRECONDITIONERS

**Lemma** (Lemma 2, restated). *For a general preconditioner $P$, for linear regression with stochastic Gaussian inputs, the following holds:*

$$\mathbb{E}[\ell^{(t)}] \leq O\left[\left(1 - \frac{\lambda_{\min}(\boldsymbol{A}(P))}{3\operatorname{Tr}(\boldsymbol{A}(P))}\right)^t \ell^{(0)}\right].$$

*Proof.* Let's define

$$M^{(t)} = \mathbb{E}[(\theta^{(t)} - \theta^\star)(\theta^{(t)} - \theta^\star)^\top]. \tag{12}$$

For any preconditioner $P$, given the update $\theta^{(t+1)} - \theta^* = (\mathbf{I} - \eta P x x^\top)(\theta^{(t)} - \theta^*)$, we have:

$$
\begin{aligned}
M^{(t+1)} &= M^{(t)} - \eta P \Sigma_x \theta^{(t)} - \eta \theta^{(t)} \Sigma_x P + \eta^2 P\left(2\Sigma_x M^{(t)} \Sigma_x + \operatorname{Tr}(\Sigma_x M^{(t)})\Sigma_x\right)P \\
&= (\mathbf{I} - \eta P \Sigma_x) M^{(t)} (\mathbf{I} - \eta P \Sigma_x)^\top + \eta^2 P\left(\Sigma_x M^{(t)} \Sigma_x + \operatorname{Tr}(\Sigma_x M^{(t)})\Sigma_x\right)P.
\end{aligned}
\tag{13}
$$

Observe that $\mathbb{E}[\ell_t] = \mathbb{E}[\operatorname{Tr}(\Sigma_x M^{(t)})]$. This motivates us to make the following definition:

$$\widetilde{M}^{(t)} = \Sigma_x^{1/2} M^{(t)} \Sigma_x^{1/2}, \tag{14}$$

and so $\mathbb{E}[\ell_t] = \mathbb{E}[\operatorname{Tr}(\widetilde{M}^{(t)})]$.

Define $\boldsymbol{A}(P) := \Sigma_x^{1/2} P \Sigma_x^{1/2}$. The corresponding update rule is then:

$$
\begin{aligned}
\widetilde{M}^{(t+1)} &= (\mathbf{I} - \eta \boldsymbol{A}(P))\widetilde{M}^{(t)}(\mathbf{I} - \eta \boldsymbol{A}(P))^\top + \eta^2 \boldsymbol{A}(P)\left(\widetilde{M}^{(t)} + \operatorname{Tr}(\widetilde{M}^{(t)})\mathbf{I}\right)\boldsymbol{A}(P) \\
&\preceq (\mathbf{I} - \eta \boldsymbol{A}(P))\widetilde{M}^{(t)}(\mathbf{I} - \eta \boldsymbol{A}(P))^\top + 2\eta^2 \operatorname{Tr}(\widetilde{M}^{(t)})(\boldsymbol{A}(P))^2.
\end{aligned}
\tag{15}
$$

For a given $P$, achieving the best loss contraction rate reduces to finding the optimal $\eta$. Rotating the left and the right hand side into the eigenbasis of $\boldsymbol{A}(P)$ and noting that the Trace of a matrix is independent of rotation, we can consider diagonal $\boldsymbol{A}(P)$ (without loss of generality) with the diagonal entries correspond to the eigenvalues. Define:

$$v_t = \operatorname{diag}(\widetilde{M}^{(t)}).$$

We have:

$$v_{t+1} = ((\mathbf{I} - \eta \boldsymbol{A}(P))^2 + \eta^2 \boldsymbol{A}(P)^2 + \eta^2 \operatorname{diag}(\boldsymbol{A}(P)^2)\vec{1}^\top)v_t. \tag{16}$$

Taking dot product with $\operatorname{diag}(\boldsymbol{A}(P)^{-1})$ on both sides, we get

$$
\begin{aligned}
\operatorname{diag}(\boldsymbol{A}(P)^{-1})^\top v_{t+1} &= \operatorname{diag}(\boldsymbol{A}(P)^{-1})^\top ((\mathbf{I} - \eta \boldsymbol{A}(P))^2 + \eta^2 \boldsymbol{A}(P)^2 + \eta^2 \operatorname{diag}(\boldsymbol{A}(P)^2)\vec{1}^\top)v_t \\
&= \operatorname{diag}(\boldsymbol{A}(P)^{-1})^\top v_t - 2\eta 1^\top v_t + 2\eta^2 \operatorname{diag}(\boldsymbol{A}(P))^\top v_t + \eta^2 \operatorname{Tr}(\boldsymbol{A}(P))1^\top v_t.
\end{aligned}
\tag{17}
$$

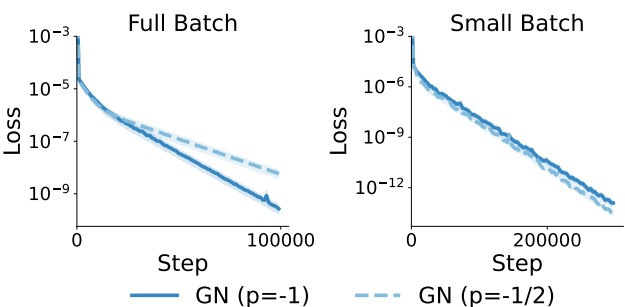

Figure 6: Comparing GN power $p \in \{-\frac{1}{2}, -1\}$. Contrary to Section 5, when $\text{GN}^{-1}$ has a more favorable condition number, it converges faster or close to $\text{GN}^{-\frac{1}{2}}$ on both small and large batches.

Since $\widetilde{M}^{(t)}$ is PSD, $v_t$ has non-negative entries. Hence we have $1^\top v_t = \text{diag}(\boldsymbol{A}(P)^{-1})^\top \boldsymbol{A}(P) v_t \geq \lambda_{\min}(\boldsymbol{A}(P))\text{diag}(\boldsymbol{A}(P)^{-1})^\top v_t$, and

$$\text{diag}(\boldsymbol{A}(P)^{-1})^\top v_{t+1}$$
$$\leq \text{diag}(\boldsymbol{A}(P)^{-1})^\top v_t - 2\eta 1^\top v_t + \lambda_{\max}(\boldsymbol{A}(P))2\eta^2 1^\top v_t + \eta^2 \text{Tr}(\boldsymbol{A}(P)1^\top v_t$$
$$= \text{diag}(\boldsymbol{A}(P)^{-1})^\top v_t - \eta \cdot \big(2 - (2\lambda_{\max}(\boldsymbol{A}(P)) + \text{Tr}(\boldsymbol{A}(P))\eta\big)1^\top v_t \tag{18}$$
$$\leq \left(1 - \lambda_{\min}(\boldsymbol{A}(P)) \cdot \eta\big(2 - (2\lambda_{\max}(\boldsymbol{A}(P)) + \text{Tr}(\boldsymbol{A}(P))\eta\big)\right) \cdot \text{diag}(\boldsymbol{A}(P)^{-1})^\top v_t.$$

The max contraction rate is achieved by setting $\eta = \frac{1}{2\lambda_{\max}(\boldsymbol{A}(P))+\text{Tr}(\boldsymbol{A}(P))}$, which gives

$$\text{diag}(\boldsymbol{A}(P)^{-1})^\top v_{t+1} \leq \left(1 - \frac{\lambda_{\min}(\boldsymbol{A}(P))}{2\lambda_{\max}(\boldsymbol{A}(P)) + \text{Tr}(\boldsymbol{A}(P))}\right)\text{diag}(\boldsymbol{A}(P)^{-1})^\top v_t$$
$$\leq \left(1 - \frac{\lambda_{\min}(\boldsymbol{A}(P))}{3\,\text{Tr}(\boldsymbol{A}(P))}\right) \cdot \text{diag}(\boldsymbol{A}(P)^{-1})^\top v_t. \tag{19}$$

$\square$

### B.3 COMPARING GN POWERS

This section discusses the comparison between $\text{GN}^{-1}$ and $\text{GN}^{-\frac{1}{2}}$. We will show that under the identity basis, $\text{GN}^{-1/2}$ can outperform $\text{GN}^{-1}$ even with full batches.

One can show that the convergence rate of preconditioned gradient descent (Boyd & Vandenberghe, 2004) depends on the condition number of the preconditioned Hessian given by

$$\kappa(\boldsymbol{A}(P)) := \frac{\lambda_{\max}(\boldsymbol{A}(P))}{\lambda_{\min}(\boldsymbol{A}(P))}. \tag{20}$$

By Lemma 2, to compare these two powers, it suffices to compare the condition number of for specific preconditioners $\Sigma_x^{-1}$ and $\Sigma_x^{-1/2}$. We claim that there exists problems for which, even in the full batch case, in identity basis, $p = 0.5$ leads to faster convergence than $p = 1$:

**Claim 1.** *There exists $\Sigma_x$ such that $\kappa(\boldsymbol{A}(diag(\Sigma_x)^{-1/2})) < \kappa(\boldsymbol{A}(diag(\Sigma_x)^{-1}))$.*

Denote $r(\Sigma_x) := \frac{\kappa(\Sigma_x^{1/2}\text{diag}(\Sigma_x^{-1})\Sigma_x^{1/2})}{\kappa(\Sigma_x^{1/2}\text{diag}(\Sigma_x^{-1/2})\Sigma_x^{1/2})}$. We empirically show that there exists $\Sigma_x$ such that $r(\Sigma_x) > 1$. We obtain such $\Sigma_x$ by fixing the diagonal matrix of eigenvalues $\Lambda$ and randomly sampling orthonormal matrices $U$, and setting $\Sigma_x = U\Lambda U^\top$.

In particular, we construct two covariance matrices $\Sigma_{\text{half}}, \Sigma_{\text{one}} \in \mathbb{R}5 \times 5$, such that $r(\Sigma_{\text{half}}) > 1$ (i.e. $\text{GN}^{-1/2}$ is more favorable), and $r(\Sigma_{\text{one}}) < 1$ (i.e. $\text{GN}^{-1}$ is more favorable). As shown in Figure 1

(right) and Figure 6, $GN^{-\frac{1}{2}}$ indeed converges faster on data from $\Sigma_{\text{half}}$, whereas $GN^{-1}$ converges faster with $\Sigma_{\text{one}}$, consistent with the theory.

Characterizing covariance matrices for which $r(\Sigma_x) > 1$ is left as future work.

## C  PROOF FOR THE LOGISTIC EXAMPLE

For logistic regression, recall that the gradient $g(\theta)$ and the diagonal Gauss–Newton matrix $H^{(\text{GN})}(\theta)$ are given by

$$[g(\theta)]_i = 2\nu_i\,\theta_i\big(\sigma(\theta_i^2) - P_i\big), \quad [H^{(\text{GN})}(\theta)]_{ii} = 4\nu_i\,\theta_i^2\,\sigma(\theta_i^2)\big(1 - \sigma(\theta_i^2)\big). \tag{21}$$

For a step size sequence $\{\eta^{(t)}\}$ and ridge regularization $\alpha \geq 0$, the Gauss-Newton iteration is

$$\theta^{(t+1)} = M_{\eta^{(t)},\alpha}(\theta^{(t)}), \qquad \text{where } M_{\eta,\alpha}(\theta) = \theta - \eta\,(H^{(\text{GN})}(\theta) + \alpha I)^{-1}g(\theta).$$

Coordinate-wise, this update reads

$$[\,M_{\eta,\alpha}(\theta)\,]_i \;=\; \theta_i - , \frac{2\theta_i\nu_i(\sigma(\theta_i^2) - P_i)}{4(\theta_i)^2\nu_i\,\sigma(\theta_i^2)(1 - \sigma(\theta_i^2)) + \alpha}. \tag{22}$$

### C.1  PROOF FOR THEOREM 2

The proof strategy hinges on the following key technical lemma that establishes a learning rate threshold for a single coordinate, where the threshold is a function of both initialization $\theta^{(0)}$ and the regularization parameter $\alpha$. Above this threshold, the Gauss-Newton update diverges. By requiring that all coordinates avoid this divergence to ensure global convergence, we use the lemma to derive a strict upper bound on the algorithm's final learning rate, $\eta^\infty$. Substituting this necessary restriction into the definition of the local contraction factor directly yields the theorem's lower bound, showing that slow convergence is an unavoidable consequence of global stability from the chosen initialization.

**Lemma 3.** *For constants $\eta, \alpha > 0$, define the one-dimensional update map $M_{\eta,\alpha} : \mathbb{R} \to \mathbb{R}$ corresponding to a regularized Gauss-Newton step:*

$$M_{\eta,\alpha}(\theta) = \theta - \eta\frac{2\theta(\sigma(\theta^2) - P)}{4\theta^2\sigma(\theta^2)(1 - \sigma(\theta^2)) + \alpha}.$$

*Consider the update rule $\theta^{(t+1)} = M_{\eta^{(t)},\alpha}(\theta^{(t)})$.*

*There exists a universal constant c such that for any target probability $P \in [0.6, 0.8]$ and any initial weight $\theta^{(0)} > 0$ satisfying $\sigma((\theta^{(0)})^2) \leq 0.55$, if the learning rate sequence satisfies*

$$\eta^{(t)} \geq c\sqrt{\log \frac{1}{\theta^{(0)}}}\left(\theta^{(0)} + \frac{\alpha}{\theta^{(0)}}\right) \quad \text{for all } t,$$

*then $|\theta^{(t)}|$ diverges geometrically.*

*Proof.* The proof proceeds in three parts. First, we show that the first step, $w^{(1)}$, becomes large. Second, we establish a key property satisfied by $w^{(1)}$. Finally, we use this to show that all subsequent iterates grow geometrically.

*Part 1: The first step makes $\theta^{(1)}$ large.* The condition $\sigma((\theta^{(0)})^2) \leq 0.55$ implies $\theta^{(0)} \leq 0.5$. Since $P \geq 0.6$, the term $\sigma((\theta^{(0)})^2) - P$ is negative, ensuring $\theta^{(1)} > \theta^{(0)}$. We can lower bound $\theta^{(1)}$ as

follows:

$$\theta^{(1)} = \theta^{(0)} - \eta^{(0)} \frac{2\theta^{(0)}(\sigma((\theta^{(0)})^2) - P)}{4(\theta^{(0)})^2\sigma((\theta^{(0)})^2)(1 - \sigma((\theta^{(0)})^2)) + \alpha}$$

$$\geq \theta^{(0)} + \eta^{(0)} \frac{2\theta^{(0)}(0.6 - 0.55)}{4(\theta^{(0)})^2(0.55)(0.45) + \alpha}$$

$$\geq \eta^{(0)} \frac{0.1\theta^{(0)}}{(\theta^{(0)})^2 + \alpha}$$

$$= \eta^{(0)} \frac{0.1}{\theta^{(0)} + \alpha/\theta^{(0)}}.$$

Substituting our lower bound for $\eta^{(0)}$ from the lemma statement yields:

$$\theta^{(1)} \geq \left( c\sqrt{\log \frac{1}{\theta^{(0)}}} \left(\theta^{(0)} + \frac{\alpha}{\theta^{(0)}}\right) \right) \frac{0.1}{\theta^{(0)} + \alpha/\theta^{(0)}} = 0.1c\sqrt{\log \frac{1}{\theta^{(0)}}}.$$

As $\theta^{(0)} \to 0$, this lower bound grows, so we can choose the universal constant $c$ large enough to make $\theta^{(1)}$ arbitrarily large.

*Part 2: Establishing a key property of $\theta^{(1)}$.* We now show we can choose $c$ large enough to ensure two conditions hold simultaneously for $\theta^{(1)}$:

  (i)  $\sigma((\theta^{(1)})^2) \geq 0.9$.

  (ii)  $4(\theta^{(1)})^2(1 - \sigma((\theta^{(1)})^2)) \leq \theta^{(0)}$.

Condition (i) is met by choosing $c$ sufficiently large. For condition (ii), we use the facts that $1 - \sigma(z) \leq e^{-z}$ and that $z(1 - \sigma(z))$ is a decreasing function for $z \geq 2$. [3] Let $\theta_{\text{low}}^{(1)} := 0.1c\sqrt{\log(1/\theta^{(0)})}$. This implies:

$$4(\theta^{(1)})^2(1 - \sigma((\theta^{(1)})^2)) \leq 4(\theta_{\text{low}}^{(1)})^2(1 - \sigma((\theta_{\text{low}}^{(1)})^2)) \leq 4(\theta_{\text{low}}^{(1)})^2 e^{-(\theta_{\text{low}}^{(1)})^2}$$

$$= 4(0.01c^2)\log(1/\theta^{(0)}) \cdot \exp\left(-(0.01c^2)\log(1/\theta^{(0)})\right)$$

$$= (0.04c^2)\log(1/\theta^{(0)}) \cdot (\theta^{(0)})^{0.01c^2}$$

$$= \left((0.04c^2)\log(1/\theta^{(0)})(\theta^{(0)})^{0.01c^2-1}\right) \cdot \theta^{(0)}.$$

For a sufficiently large constant $c$ (e.g., $0.01c^2 > 2$), the term in the large parenthesis is less than 1, because for a fixed $\theta^{(0)} \in (0, 0.5]$, the polynomial term $(\theta^{(0)})^{0.01c^2-1}$ decays much faster than the logarithmic term $\log(1/\theta^{(0)})$ grows. This establishes condition (ii).

*Part 3: Proving geometric divergence for $t \geq 1$.* Consider any $\eta$ satisfying the learning rate lower bound, i.e. suppose:

$$\eta \geq c\sqrt{\log \frac{1}{\theta^{(0)}}} \left(\theta^{(0)} + \frac{\alpha}{\theta^{(0)}}\right).$$

We show that for any $\theta$ where $\theta^2 \geq (\theta^{(1)})^2$, it follows that $|M_{\eta,\alpha}(\theta)| \geq \sqrt{2}|\theta|$. First, rewrite the update as

$$M_{\eta,\alpha}(\theta) = \theta\left(1 - \frac{2\eta(\sigma(\theta^2) - P)}{4\theta^2\sigma(\theta^2)(1 - \sigma(\theta^2)) + \alpha}\right).$$

Let $K(\theta) = \frac{2\eta(\sigma(\theta^2) - P)}{4\theta^2\sigma(\theta^2)(1 - \sigma(\theta^2)) + \alpha}$. We seek to show $|1 - K(\theta)| \geq \sqrt{2}$, which is true if $K(\theta) \geq 1 + \sqrt{2}$.

We lower bound $K(\theta)$ for any $\theta$ where $\theta^2 \geq (\theta^{(1)})^2$. The numerator is positive and lower-bounded using condition (i): $2\eta(\sigma(\theta^2) - P) \geq 2\eta(\sigma((\theta^{(1)})^2) - P) \geq 2\eta(0.9 - 0.8) = 0.2\eta$. For the

---

[3]Indeed, $\frac{d}{dz}[z(1 - \sigma(z))] = (1 - \sigma(z)) - z\sigma(z)(1 - \sigma(z)) < 0$ once $z \geq 2$.

denominator, we use the fact that $z(1 - \sigma(z))$ is decreasing (for $z \geq 2$), condition (ii), and that $w^{(0)} \leq 0.5$:

$$4\theta^2\sigma(\theta^2)(1-\sigma(\theta^2))+\alpha \leq 4\theta^2(1-\sigma(\theta^2))+\alpha \leq 4(\theta^{(1)})^2\big(1-\sigma((\theta^{(1)})^2)\big)+\alpha \leq \theta^{(0)}+\alpha \leq \theta^{(0)}+\frac{\alpha}{\theta^{(0)}}.$$

Combining these bounds gives:

$$K(\theta) \geq \frac{0.2\eta}{\theta^{(0)} + \alpha/\theta^{(0)}} \geq 0.2c\sqrt{\log\frac{1}{\theta^{(0)}}},$$

where the last step follows by substituting the lower bound for $\eta$. Since $\theta^{(0)} \leq 0.5$, we have $\log(1/\theta^{(0)}) \geq \log(2)$. We can choose the universal constant $c$ large enough such that $0.2c\sqrt{\log 2} \geq 1 + \sqrt{2}$.

Thus, for any $t \geq 1$, we have $|\theta^{(t+1)}| = |M_{\eta^{(t)},\alpha}(\theta^{(t)})| \geq \sqrt{2}|\theta^{(t)}|$, which shows that $|\theta^{(t)}|$ diverges geometrically. $\qquad\square$

We are now ready to complete the proof of Theorem 2.

*Proof.* (Theorem 2) The proof proceeds by using Lemma 3 to find an upper bound on the final learning rate $\eta^{(\infty)}$, and then substituting this bound into the definition of the local contraction factor $\gamma$.

At the optimum $\theta^*$, the diagonal entries of the Fisher matrix $H_*^{(\mathrm{GN})} = H^{(\mathrm{GN})}(\theta^*)$ are

$$\lambda_i(H_*^{(\mathrm{GN})}) = 4(\theta_i^*)^2\nu_i\sigma((\theta_i^*)^2)\big(1 - \sigma((\theta_i^*)^2)\big).$$

Recall the target probabilities $P_i = \sigma((\theta_i^*)^2)$ are assumed to lie in $[0.6, 0.8]$. This implies that $(\theta_i^*)^2$'s are bounded by a universal constant. Thus, the term $4(\theta_i^*)^2\sigma((\theta_i^*)^2)\big(1 - \sigma((\theta_i^*)^2)\big)$ is also bounded by universal constants, and we conclude that the Fisher eigenvalues are proportional to the sampling probabilities. In particular,

$$\lambda_{\min}(H_*^{(\mathrm{GN})}) \geq \nu_{\min}/c_1,$$

where $c_1$ is a universal constant.

The Gauss-Newton update for each coordinate $\theta_i$ can be analyzed independently. For the sequence $\theta^{(t)}$ to converge $\theta^*$, the iterates for each coordinate $\theta_i[t]$ must also converge to $\theta_i^*$. From equation 22, the update rule for each coordinate can be written as:

$$[\,M_{\eta,\alpha}(\theta)\,]_i \;=\; \theta_i - \eta\,\frac{2\theta_i(\sigma(\theta_i^2) - P_i)}{4(\theta_i)^2\,\sigma(\theta_i^2)(1 - \sigma(\theta_i^2)) + \alpha/\nu_i}.$$

This shows that we can apply Lemma 3 coordinate wise, where we take $\alpha/\nu_i$ as the regularization parameter. Since $\{\eta^{(t)}\}$ is non-increasing, Lemma 3 implies the limiting stepsize $\eta^{(\infty)}$ must satisfy the following for each coordinate $i \in [d]$:

$$\eta^{(\infty)} \leq c\sqrt{\log\frac{1}{\theta_i[0]}}\left(\theta_i[0] + \frac{\alpha/\nu_i}{\theta_i[0]}\right).$$

In our setting, the initial weights are $\theta_i[0] = 1/\sqrt{d}$. To get a single upper bound on $\eta^{(\infty)}$, we take the tightest possible constraint derived from above, which is when $\nu_i$ is at its maximum, $\nu_{\max}$. Thus, for convergence to be possible, $\eta^{(\infty)}$ must be bounded by:

$$\eta^{(\infty)} \leq c\sqrt{\log d}\left(\frac{1}{\sqrt{d}} + \frac{\sqrt{d}\alpha}{\nu_{\max}}\right).$$

The local contraction factor is the spectral radius of $I - \eta^{(\infty)}(H_*^{(\mathrm{GN})} + \alpha I)^{-1}H_*^{(\mathrm{GN})}$. Its eigenvalues are $1 - \eta^{(\infty)}\frac{\lambda_i(H_*^{(\mathrm{GN})})}{\lambda_i(H_*^{(\mathrm{GN})})+\alpha}$. We can lower bound the spectral radius by considering the smallest

eigenvalue of $H_*^{(\text{GN})}$, $\lambda_{\min}$:

$$\gamma(\eta^\infty, \lambda) \geq 1 - \eta^\infty \frac{\lambda_{\min}(H_*^{(\text{GN})})}{\lambda_{\min}(H_*^{(\text{GN})}) + \lambda}.$$

Substituting the upper bound on $\eta^{(\infty)}$ from above:

$$\gamma(\eta^{(\infty)}, \alpha) \geq 1 - \left( c\sqrt{\log d} \left( \frac{1}{\sqrt{dim}} + \frac{\sqrt{dim}\alpha}{\nu_{\max}} \right) \right) \frac{\lambda_{\min}}{\lambda_{\min} + \alpha}$$

$$= 1 - c\sqrt{\log d} \left( \frac{1}{\sqrt{d}} \cdot \frac{\lambda_{\min}}{\lambda_{\min} + \alpha} + \frac{\sqrt{d}\alpha}{\nu_{\max}} \cdot \frac{\lambda_{\min}}{\lambda_{\min} + \alpha} \right).$$

We can bound the terms in the parenthesis using $\frac{\lambda_{\min}}{\lambda_{\min} + \alpha} \leq 1$ and $\frac{\lambda_{\min}}{\lambda_{\min} + \alpha} \leq \frac{\lambda_{\min}}{\lambda_{\min} + \alpha}$:

$$\gamma(\eta^{(\infty)}, \alpha) \geq 1 - c\sqrt{\log d} \left( \frac{1}{\sqrt{d}} + \frac{\sqrt{d}\lambda_{\min}}{\nu_{\max}} \right).$$

Using the our lower bound $\lambda_{\min} \geq \nu_{\min}/c_1$ from above, and absorbing constants into $c'$:

$$\gamma(\eta^{(\infty)}, \alpha) \geq 1 - c'\sqrt{\log d} \left( \frac{1}{\sqrt{d}} + \sqrt{d}\frac{\nu_{\min}}{\nu_{\max}} \right),$$

which implies the claimed result. $\qquad\square$

# D  EXPERIMENTS

## D.1  ADDITIONAL EXPERIMENT INFORMATION

**Hyperparameters, hardware, and runtime**   The learning rate ($\eta$) search is first performed within $[10^{-5}, 0.1]$ at factors of 3 (e.g. $0.01, 0.003, 0.001$) and then at factors of 2 or finer around the optimal value. The regularization ($\alpha$) search is at factors at 10 (e.g. $10^{-3}, 10^{-4}$). The optimal values typically lie within $10^{-2}$ to $10^{-6}$, and $\text{GN}^{-1}$ tends to require larger $\alpha$ than $\text{GN}^{-1/2}$. For Adam, we additionally sweep over $\beta_2 \in \{0, 0.9, 0.95, 0.99\}$. When comparing batch sizes, we vary the batch size used for computing gradient, and always use a large batch size (4096) for Gauss-Newton matrix to ensure an accurate basis estimation. Experiments were run on NVIDIA A100 GPUs. Simulation runs in Section 3.1.2 each completes within 1min. Simulation runs in Section 3.3 takes 9min for every 100k steps. The parity and staircase runs take around 10min for every 1k steps. For CIFAR experiments, runs under the identity basis take less than 5min each, and runs under the Kronecker approximation of the eigenbasis take around 80min each.

**Kronecker factorization**   Inspired by prior work (Martens & Grosse, 2015; Gupta et al., 2018; Vyas et al., 2024), our experiments use Kronecker factorization as a computationally efficient approximation to the full eigenbasis. Given a matrix-valued parameter $W \in \mathbb{R}^{m \times n}$, let $g \in \mathbb{R}^{mn}$ denote the flattened gradient, and $G \in \mathbb{R}^{m \times n}$ denote the unflattened gradient. The $mn \times mn$ Gauss-Newton matrix can be approximated by a Kronecker factorization as

$$H^{(\text{GN})} := \mathbb{E}[gg^\top] \approx \mathbb{E}[GG^\top] \otimes \mathbb{E}[G^\top G], \tag{23}$$

where $\otimes$ denote the Kronecker product. The eigenvalues and eigenvectors of the Kronecker product are the products and Kronecker products of the factors; hence the eigenbasis of $H^{(\text{GN})} \in \mathbb{R}^{mn \otimes mn}$ can be approximated by computing the eigenbasis of the smaller $\mathbb{E}[GG^\top] \in \mathbb{R}^{m \times m}$, $\mathbb{E}[G^\top G] \in \mathbb{R}^{n \times n}$.

*Is Kronecker approximation a good proxy for the full eigenbasis?* Benzing (2022) showed that Kronecker-factored approximation such as KFAC (Martens & Grosse, 2015) can lead to better performance than using the full eigenbasis in some cases. They attributed the gain to heuristic damping, which effectively controls the step sizes and was beneficial in their experiments. Our experiments do not use such heuristic damping, and we find the Kronecker approximation to behave similarly to the full eigenbasis, while being much more compute-efficient.

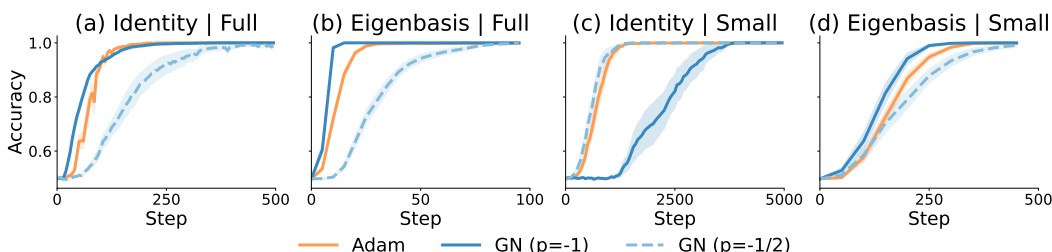

Figure 7: **Sparse parity**, comparing Adam, $GN^{-1}$ and $GN^{-1/2}$ for the full $2 \times 2$ grid (Table 1).

D.2    DETAILS FOR MLP EXPERIMENTS WITH SQUARED LOSS

This section provides details for the MLP experiments in Section 5.

We learn all tasks with single-hidden-layer MLPs given by $\hat{y} = f(x; \theta) = a \cdot \sigma(w^\top x + b)$, where $w \in \mathbb{R}^{d \times m}, a, b \in \mathbb{R}^d$, with $d$ and $m$ being the input and hidden dimensions respectively. The non-linearity $\sigma$ defaults to ReLU unless specified otherwise.

Below we provide detailed descriptions of the tasks:

- **Learning from a random teacher network.** We first construct a teacher-student setting to evaluate our hypotheses on a non-convex example. Consider input vectors, $\mathbf{x}_i \sim \mathcal{N}(0, \Sigma_x) \in \mathbb{R}^d$ where $\Sigma_x \in \mathbb{R}^{d \times d}$ is a random covariance matrix. We initialize a random teacher model, $f_\mathcal{T} : \mathbb{R}^d \to \mathbb{R}$, as a single hidden layer MLP. For each input vector, we sample output labels from the teacher: $y_i = f(x_i; \theta_\mathcal{T}) = a_\mathcal{T} \cdot \sigma(w_\mathcal{T}^\top x_i + b_\mathcal{T})$, where, $w_\mathcal{T} \in \mathbb{R}^{m \times d}; a_\mathcal{T}, b_\mathcal{T} \in \mathbb{R}^{\rhd}$, and $n_\mathcal{T}, h_\mathcal{T}$ are the teacher's input and hidden dimensions, respectively. The objective is to learn this $(x, y)$ mapping using an identical student model which has a hidden dimension $d_\mathcal{S} = 2 \times d_\mathcal{T}$.

- **Feature learning with sparse parity.** Sparse parity is well-studied and widely adopted for understanding neural network optimization (Barak et al., 2022; Bhattamishra et al., 2022; Edelman et al., 2023; Morwani et al., 2023; Abbe et al., 2024). It can be viewed as learning a sparse "feature" embedded in a much higher ambient dimension. Specifically, $(d, k)$-parity is a function from $\boldsymbol{x} \in \{\pm 1\}^d$ to $y = \prod_{i \in \mathcal{S}} x_i \in \{\pm 1\}$, where $\mathcal{S} = \{s_1, s_2, \ldots, s_k\} \subseteq [d]$ is the unknown support of relevant coordinates. In our experiments, we set $d = 20, k = 6$.

- **Feature learning with staircase.** We consider a multi-feature generalization of sparse parity called the staircase function (Abbe et al., 2022; 2023). Given input $x \in \{\pm 1\}^d$, the label $y$ is the sum of several parity functions, whose supports are specified by $k$ segments. Specifically, $y = \sum_{(s_i, e_i) \in \mathcal{P}} \prod_{j=s_i}^{e_i - 1} x_j$, where $\mathcal{P} = \{(s_i, e_i)\}_{i \in [k]}$ are the start (inclusive) and stop (exclusive) indices of a segment. For our experiments, we set each segment to be of the same size and choose $d = 21, k = 3$, i.e., $\mathcal{P} = \{(0, 7), (7, 14), (14, 21)\}$, and $y \in \{-3, -1, 1, 3\}$.

- **CIFAR-10** (Krizhevsky et al., 2009). The input images are flattened to a length-3072 vector and the labels are treated as 10-dimensional one-hot vectors. We use 400 steps in all experiments, which is sufficient for large-batch eigenbasis experiments to reach around 47% accuracy, a reasonable performance for 2-layer MLPs.

*What about using power $p = -1$ for Adam?* In Section 2, we introduced the power $p \in \{-\frac{1}{2}, -1\}$ as a hyperparameter for Gauss-Newton (GN) but kept the power Adam to be $-\frac{1}{2}$, following the standard definition of the Adam algorithm. For completeness, we experiment on Adam with $p = -1$ on sparse parity. Our results in Figure 8 is consistent with Lin et al. (2024), which finds that $p = -1$ shows comparable empirical performance to the standard choice of $p = -0.5$, especially under low-precision.

D.3    INTERPOLATING BETWEEN BASIS

In Section 5, we discussed the behavior of GN and Adam on two kinds of basis: identity and full-GN, depicting an incorrect and a correct basis to precondition the gradient, respectively. To provide a

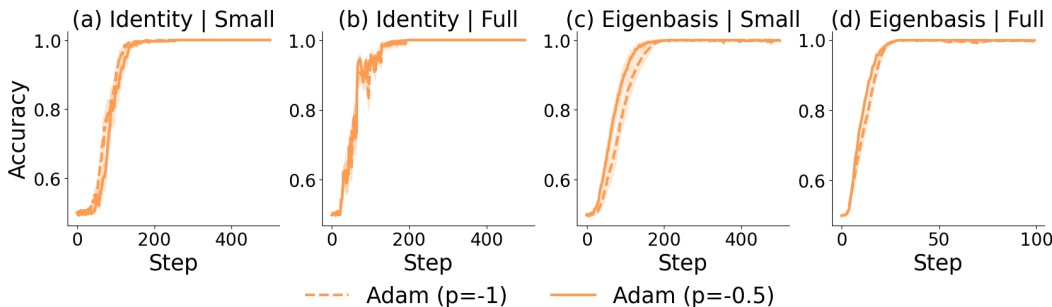

Figure 8: **Sparse parity**, comparing Adam, with power $-1$ or $-\frac{1}{2}$ for the full $2 \times 2$ grid (Table 1).

---

**Algorithm 2** Geodesic interpolation between bases

---
17:   **Input:** full GN basis $U$, interpolation factor $\alpha$.
18:     Compute the matrix log $K := \mathrm{logm}(U)$.
19:     Compute the matrix exponent $\hat{U} := \exp(\alpha \cdot K)$.
20:     Obtain the real part $U_\alpha := \mathrm{real}(\hat{U})$.
21:   **Output:** $U_\alpha$.

---

more complete picture of the effect of basis, we provide results with more granularity with respect to the choice of basis.

Particularly, we compare GN and Adam on bases of "intermediate" quality by interpolating between the identity and full-GN basis. Given the identity basis $I$ and the eigenbasis $U$, we construct an interpolation $U_\gamma$, parameterized by some interpolation factor $\gamma \in \{0, 0.25, 0.5, 0.75, 1\}$, using geodesic interpolation (Algorithm 2). In particular, $U_0 = I$ and $U_1 = U$. Results are shown in Figure 9. In particular, Adam and $\mathrm{GN}^{-\frac{1}{2}}$ behave similarly under the stochastic regime across basis choices, as predicted by the theory (Section 3.2).

### D.4   DETAILS FOR LOGISTIC EXPERIMENTS

**Simulation for Section 4**   We run simulation following the 2-layer linear network example in Section 4. The inputs are 2048-dimensional one-hot vectors following a power law decay, with $\nu_i := \mathbf{Pr}(x = e_i) \propto i^{-c}$. We set $c = 0.6$ in the experiments. The label distributions are set to $P_i = p(y = 1 | x = e_i) = 0.75$ for all $i$.

**Transformer experiments**   This section provides details for the Transformer experiments in Section 5. The attention module shares a similar structure as the logistic regression results in Section 4: the inner product of query and key matrices resembles the reparameterization, and the softmax function resembles the logistic function. As a result, Gauss-Newton is forced to take conservative step sizes and suffers from slow convergence, as shown in Figure 5.

We consider a selection task: The input is a sequence of $T$ Gaussian vectors followed by a length-$d$ one-hot vector ($d \geq T$) specifying which input is used in the regression task, i.e. $[x_1, \cdots, x_T, s]$. For $s = e_i$, the label is given by $y = \langle \theta_*, x_i \rangle$. We set $T = 32$ in the experiments. The task is learned with a 1-layer 1-head Transformer with dimension 128. Figure 5 shows results comparing GN and Adam under the Kronecker-approximated eigenbasis with large batches (batch size = 16384), each aggregated over 10 seeds. Each run takes around 90min to complete.

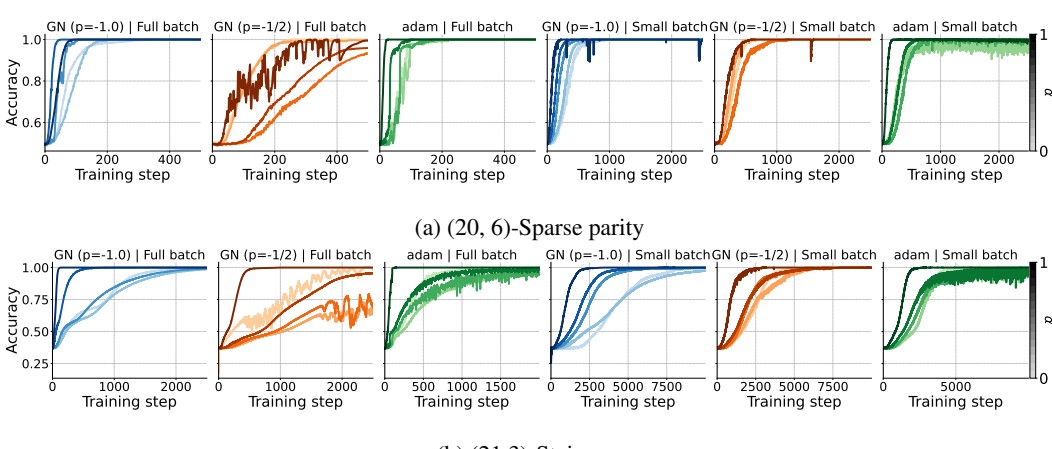

(a) (20, 6)-Sparse parity

(b) (21,3)-Staircase

Figure 9: **Basis interpolation**: Comparing $GN^{-1}$, $GN^{-1/2}$, and Adam under various bases, for parity and staircase (Section 5). Each basis is obtained by a geometric interpolation between the eigenbasis (darker colors) and the identity basis (lighter colors), parmaeterized by a factor $\gamma \in \{0, 0.25, 0.5, 0.75, 1\}$.

