# OpenReview forum: "Adam or Gauss-Newton? — A Comparative Study In Terms of Basis Alignment and SGD Noise"
_ICLR.cc/2026/Conference — Submitted to ICLR 2026_

### Official Review · Reviewer_sdk6 · 2025-10-31

**Soundness:** 2
**Presentation:** 3
**Contribution:** 2
**Rating:** 4
**Confidence:** 3

**Summary:**

This paper compares two optimization methods, Adam and Gauss Newton (GN) method, decomposes the parameter update formula in the preconditioned optimizer, and establishes an analytical framework based on the following two aspects: the choice of basis in the preconditioner, and the impact of gradient noise from mini-batching. This helps to gain a deeper understanding of optimization algorithms such as Adam and Gauss Newton methods.

**Strengths:**

This paper provides some observations and  conclusions, such as:
	1.GN is optimal under the ideal eigenbasis. In the correct basis, $GN^{-1}$ is the optimal preconditioner, in both full-batch and the stochastic regime.
	2. GN is sensitive to the basis choice, when the basis is misaligned, Adam can outperform both , $GN^{-1}$ and , $GN^{-1/2}$.
This paper provides theoretical analysis and proof of the above conclusion based on two examples: linear regression and logistic regression.

**Weaknesses:**

1.The examples of linear regression and logistic regression are too simplistic, and the optimization algorithm is well studied for the simple linear models. This paper contains relatively little theoretical analysis on the more commonly used MLP and attention, and is far from the recent architectures of neural network.
2. The analyses of this paper are based on several assumptions which is not well/further justified. E.g., It assumes the input is Gasussian, which may obtained by linear model, but it is difficult to obtain layer-wise Gaussian input in DNNs. This assumption limits the analyses extending to DNNs. Besides, this paper assume that “the gradient norms are the same for coordinate within the same block”, why is that? Can provide more illustration?
3.This paper does not analyze whether any optimizer can analyze from the perspectives of "the choice of basis and the impact of grade noise", nor does it provide proof.
4.The description of hyperparameter tuning (such as learning rate and regularization coefficient) in the experiment is relatively brief. Suggest providing a more detailed hyperparameter table or search scope in the appendix.

**Questions:**

See weaknesses.

---

> ### Author Response · Authors · 2025-11-18
>
> We thank the reviewer for their helpful feedback! We'd like to clarify the broader implications of our results.
>
> - **Connection of linear regression and logistic regression to practice**: the reviewer is concerned that these settings are much simpler compared to modern architectures, and that they have been well-studied. While we agree with the reviewer, we’d like argue that there are still many unanswered questions for these seemingly simplistic settings that can be informative to practice [1,2], and it’s important to understand these properly before moving to more complex architectures.
>
>     Based on our current empirical results, we hypothesize that MLP experiments generally agree with the linear regression results, and Transformer experiments, due to the softmax function in attention, likely align better with logistic regression where Adam can outperform Gauss-Newton even under the identity basis. The actual comparison for a particular practical setup will depend on the exact data and architecture choice and is an interesting future direction.
>     Moreover, the auto-tuning effect of Adam in Sec 3.1.2 is a concrete takeaway that we believe to be generically applicable. The auto-tuning effect, combined with Corollary 1 showing that such an effect does not exist at small batch sizes, suggests that using the full-batch gradient as preconditioner could be useful at small batch sizes, which is consistent with a concurrent work [3].
>
>     [1] Wu et al. 2025: Risk Comparisons in Linear Regression: Implicit Regularization Dominates Explicit Regularization.
>
>     [2] Wu et al. 2025: Large Stepsizes Accelerate Gradient Descent for Regularized Logistic Regression.
>
>     [3] Roulet and Agarwala 2025: Per-example gradients: a new frontier for understanding and improving optimizers.
>
> - **Whether the proposed disentanglement applies to other optimizers**: We believe our framework of “disentangling the choice of basis and the choice of diagonal preconditioner within the basis” is generally applicable, and we advocate for taking this disentangled view when comparing preconditioned optimizers. Though note that the exact comparison will be optimizer-dependent. Our paper focus on the comparison between Adam and Gauss-Newton, and do not claim beyond that. Given the prevalence use of Adam and GN optimizers, a careful comparison among the two is relevant and timely.
>
> Some other clarifications:
>
> - **Why the gradient norms are the same for coordinates within the same block**: sorry for the confusion: this is not an assumption (we have updated the wording in the revision), but rather comes from the symmetry in the problem: in Sec 3.1.2, we are considering updates with population gradients of the form $\Sigma_x (\theta - \theta^*)$, hence:
>     - Coordinates within the block of $11^\top$get exactly the same gradient (i.e. with norm $\|g_0\|$).
>     - Coordinates within the block of $I$ get gradient are of the same order, though not necessarily exactly the same (i.e. as denoted by $g_i$ for $i \in [d]$).
>
>     We have updated the paper (revisions highlighted in blue) to make this clearer.
>
> - **A more detailed report on the hyperparameters**: We have updated the appendix to clarify the range of hyperparameter search.
>
>
> Please let us know if you have other comments or suggestions, and thanks again for your review!

---

### Official Review · Reviewer_UWDe · 2025-10-31

**Soundness:** 2
**Presentation:** 2
**Contribution:** 3
**Rating:** 4
**Confidence:** 3

**Summary:**

This paper provides a comparative analysis of Adam and Gauss-Newton (GN) optimizers by evaluating them along two key axes: (1) the choice of preconditioning basis (identity vs. the GN eigenbasis) (2) the level of gradient noise (full-batch vs. small-batch) .

The work's main findings are twofold. First, in the full-batch (low-noise) setting, Adam can surprisingly outperform GN. The authors provide a quadratic example where Adam "auto-tunes" to the curvature in the identity basis, while GN fails and performs like simple gradient descent. They also present a logistic regression example where Adam converges faster than $GN^{-1}$ even in the ideal eigenbasis. Second, in the stochastic (high-noise) setting, the authors demonstrate a strong connection between Adam and $GN^{-1/2}$. They theoretically show for linear regression that Adam's preconditioner becomes approximately equivalent to $GN^{-1/2}$, regardless of the basis. This suggests that Adam's empirical design is well-suited for the high-noise regime, effectively mirroring a square-root curvature preconditioner. These theoretical insights are supported by experiments on both convex and non-convex problems.

**Strengths:**

1. The paper tackles a a critical and relevant problem: understanding the precise relationship and connection between first-order adaptive methods (like Adam) and approximate second-order methods (like Gauss-Newton). This investigation is highly relevant, especially as the choice of the preconditioning exponent (\$p\$) remains an open and poorly understood question in popular modern optimizers such as Shampoo.

2. The central contribution of proposing a \$2 \times 2\$ analytical grid—decoupling the optimizer's behavior along the axes of Basis Choice and Gradient Noise (batch size)—is excellent . This "disentanglement" is a novel and powerful lens, offering a much more granular and insightful model for comparing optimizer performance than prior work.

**Weaknesses:**

1. The paper's analysis does not fully deliver on its promise of "disentanglement". According to Algorithm 1, the authors compare Adam, which is based on a diagonal empirical Fisher preconditioner , against GN methods, which are defined using the true Gauss-Newton(GN) matrix (i.e., true Fisher). This compares two things that differ in more ways than just "basis" and "noise"—it also compares a diagonal approximation to a full-matrix AdaGrad method and an empirical statistic to a true curvature matrix. A much cleaner analysis to isolate the effect of noise would have been to compare a full-matrix empirical Fisher method (e.g., full-matrix Adagrad/Adam) against the full-matrix GN method.

2. The key theoretical argument that Adam can outperform GN rests on a single, specific counter-example: a quadratic problem where GN is forced to use the identity basis . This is a extreme argument, as no practical application of GN would use the identity matrix as its preconditioning basis. A far more common and realistic "misaligned basis" scenario would be using the basis of the empirical Fisher Information Matrix(FIM) to approximate the true GN matrix. Therefore, the conclusion that "GN converges slowly"  is not sufficiently proven; it is only shown to be true in an unrealistic, extreme case.

3. The empirical support for the paper's broad claims is very limited. The experiments are confined to simple 1-hidden-layer MLPs and a 1-layer Transformer. Whether these findings can be generalized to deep, complex, and high-dimensional architectures (e.g., modern LLMs, large CNNs) is uncertain, because the differences between these optimizers are most critical in these architectures.

**Questions:**

1. Could you please clarify the practical and theoretical motivation for studying \$GN^{-1/2}\$? In Algorithm 1, \$H^{(GN)}\$ is defined as the true Fisher matrix (approximated with a separate batch, \$X_H\$). While \$GN^{-1}\$ is the well-understood Natural Gradient Descent(NGD), the justification for applying an inverse square root (\$p=-1/2\$) to the true Fisher is unclear. Why is this specific variant a meaningful object of study?

2. Following on Weakness 2, could you extend your analysis of basis misalignment to a more practical scenario? Specifically, what would your theory predict about the performance of \$GN^{-1}\$ if it were applied in the (misaligned) eigenbasis of the empirical Fisher information matrix, rather than the identity matrix?

3. There appears to be a contradiction between your theoretical claims and your empirical results. In Figure 2(c) (Identity | Small), your plot shows \$GN^{-1}\$ (solid blue line) converging to a lower loss than Adam (orange line). This seems to contradict the claim from that Adam's "auto-tuning" allows it to outperform GN in the identity basis. Could you please explain this discrepancy?

4. Given the simplicity of the models used, what is your perspective on how these findings would apply to much deeper, state-of-the-art architectures? Do you have preliminary evidence or a theoretical argument to suggest that these conclusions will hold in more complex, practical settings?

---

> ### Author Response · Authors · 2025-11-18
>
> We thank the reviewer for finding our results novel and relevant, and for the helpful suggestions!
>
> - **There might be a misunderstanding for Weakness 1**: by Adam or Gauss-Newton, we refer to the *diagonal* preconditioner choices under a given basis (eq (2), (3) in Sec 2), rather than as conventional references to the full optimizers. The basis could be the identity, in which case both Adam and GN use a diagonal approximation (of the empirical or true Fisher); or the eigenbasis of the GN matrix, in which case both Adam and GN use information of the full GN matrix. For a given basis choice, we additionally consider whether the gradient is estimated using a full (large) or a small batch (in contrast, the GN matrix is always estimated with a large batch).
> We’d also like to note that the empirical Fisher and the true Fisher do not differ much in their basis, both provably for linear regression (as shown in our Lem 1) and empirically in more general setups (as shown in [1]).
>     [1]: Morwani et al. 24: A New Perspective on Shampoo’s Preconditioner
>
> - **GN is not generally worse than Adam:** We would like to clarify that 1) we are comparing Adam vs GN as choices of diagonal preconditioners under a given basis, and 2) we are not claiming that Adam is always better than GN on the identity basis. The construction is intended as a proof for the existence of an example, which highlights the auto-tuning effect which could be more generally interesting.
>     For more general “misaligned basis”, we provide basis interpolation results in Figure 9, where Adam outperforms GN under some interpolation of the eigenbasis and the identity basis.
>
> - **Implication to deeper, state-of-the-art architectures**: Based on our current empirical results, we hypothesize that MLP experiments generally agree with the linear regression results, and Transformer experiments, due to the softmax function in attention, likely align better with logistic regression where Adam can outperform Gauss-Newton even under the right eigenbasis. The actual comparison for a particular practical setup will depend on the exact data and architecture choice as the reviewer also alluded to, and we agree that it would be an interesting future direction.
> Moreover, the auto-tuning effect of Adam in Sec 3.1.2 is a concrete takeaway that we believe to be generically applicable. The auto-tuning effect, combined with Corollary 1 showing that such an effect does not exist at small batch sizes, suggests that using the full-batch gradient as preconditioner could be useful at small batch sizes, which is consistent with a concurrent work [2].
> [2] Roulet and Agarwala 2025: Per-example gradients: a new frontier for understanding and improving optimizers.
>
>
> Some other clarifications:
>
> - **Why studying $\text{GN}^{-1/2}$**: $\text{GN}^{-1/2}$ is of interest as the power -1/2 (of the empirical Fisher) has been adopted in AdaGrad and subsequently practical optimizers such as Shampoo, and can be preferable for stability especially with small batches.
> - **Adam performing worse than GN in the identity basis with small batches (Fig 2(c))**: We’d like to clarity that auto-tuning only occurs in the full batch setting. When using small batches, Adam’s denominator can be dominated by the gradient variance rather than gradient norm, hence voiding the auto-tuning effect which comes from the gradient norm. Thank you for raising this question, we have clarified it in the revision.
>
> Please let us know if you have other comments or suggestions, and thanks again for your review!

---

### Official Review · Reviewer_mAGy · 2025-10-31

**Soundness:** 4
**Presentation:** 3
**Contribution:** 3
**Rating:** 8
**Confidence:** 4

**Summary:**

This paper studies the relative benefit and shortcomings of the related methods of Adam-style preconditioning versus Gauss-Newton preconditioning. In particular, two additional design axes are considered: 1. it is observed that given an orthogonal basis other than the standard one, one can compare "diagonal" preconditioning with respect to differing bases, for example the basis described by the Gauss-Newton preconditioner, 2. as Adam-preconditioning takes the "square-root" power, both the inverse Gauss-Newton and inverse-square-root Gauss-Newton are taken into consideration. It is then demonstrated in two exemplar simple models of Linear Regression and Logistic Regression that the hierarchy of Adam, inverse Gauss-Newton, and inverse-sqrt Gauss-Newton shifts around significantly depending on setting, for example: preconditioning in {standard, GN eigenbasis}, and {full-batch, stochastic} updates. This clarifies the understanding that, though Adam (or Adagrad) preconditioning is related to Gauss-Newton preconditioning, the latter does not subsume the benefits of the former. Numerical experiments are provided to back up the theoretical claims, and some extrapolations are demonstrated in non-convex settings with shallow neural networks.

**Strengths:**

Overall, this is a well-written paper with clearly delineated messages and claims, and thus is worthy of acceptance. I think the overall message that there is a lot more subtlety to (Adam/Gauss-Newton) preconditioning than one might expect is well-appreciated and timely, given the many recent research efforts focused on deriving or explaining different preconditioning methods. Some theory results worth highlighting are:

1. Demonstrating how Adam "auto-tunes" to the curvature when both Adam and Gauss-Newton are provided poor preconditioning bases in a linear regression setting. This was a bit surprising, as this shows when diagonal Gauss-Newton trivially does not outcompete gradient descent, Adam when provided the same (ill-specified) basis can achieve faster convergence rates by adapting to the gradient norm. The stark difference between Adam and Gauss-Newton in this setting is surprising at first glance.

2. The local convergence vs. step-size tradeoff of Gauss-Newton in logistic regression. This result demonstrates that full Gauss-Newton, even on the correct eigenbasis, experiences tension between larger learning rates and local convergence speed, leading to overall suboptimal descent compared to even vanilla Adam. This is also surprising at first glance, considering full Gauss-Newton is a p.s.d. approximant to the Hessian.

**Weaknesses:**

Thee are no glaring deficiencies of the paper that I found. Overall, the theoretical constructions are not necessarily novel, but I think they clearly serve the purpose of identifying mismatches between Adam and Gauss-Newton. More relevantly, the theory results, while providing some suggestive intuition, do not extend to neural network settings, where it is not as clear to me how approximations to Gauss-Newton experience pathologies compared to Adam, beyond numerical demonstrations. The authors might also find the following theory paper [1] interesting, where a Kronecker-Factored approximation to the Gauss-Newton matrix seems to be a demonstrably good thing to do in some square-loss two-layer network settings.

[1] Zhang et al. "On The Concurrence of Layer-wise Preconditioning Methods and Provable Feature Learning"

**Questions:**

I have no pressing questions currently. As a curiosity, I'm wondering if the authors have intuition for whether there are generic tasks with deep neural networks such that a layer-wise factored basis (e.g. standard->diagonal, or Kronecker-Factored) is preferable to the full Gauss-Newton one. I ask because empirical evidence seems to repeatedly show the "off-diagonal blocks" have non-negligible magnitude, which layer-wise methods throw away, which suggests the factored vs full eigenbases are appreciably different.

---

> ### Author Response · Authors · 2025-11-18
>
> We thank the reviewer for their detailed review and encouraging comments!
>
> - **Zhang et al. 25 and the benefit of K-FAC**: Thank you for the reference! This paper indeed shares similar motivation as ours, such as studying whether a computationally-motivated approximation can be statistically preferable. In addition to the 2-layer neural network analyses, we also find the feature learning setting very interesting. We’ve updated the paper (revisions highlighted in blue) to include this as an interesting future direction.
>
> - **Full vs layer-wise for Gauss-Newton**: this is a very interesting question that we were wondering too, and we share a similar view as the reviewer that off-diagonal blocks may not be too detrimental — though they still have some effect, as shown in a recent work [1] on language models. For Transformers, we are currently not aware of results that where layer-wise GN is better (note that K-FAC results such as Benzing 22 and Zhang et al. 25 are for non-Transformers), and agree that understanding when this is true is an interesting direction.
> [1] Abreu et al. 25: The Potential of Second-Order Optimization for LLMs: A Study with Full Gauss-Newton.
>
> Please let us know if you have other comments or suggestions, and thanks again for your review!

---

### Official Review · Reviewer_VcjS · 2025-11-01

**Soundness:** 2
**Presentation:** 2
**Contribution:** 2
**Rating:** 4
**Confidence:** 3

**Summary:**

This paper studies the comparison between the Gauss-Newton algorithm and Adam from some theoretical perspective. The paper considers some examples and analysis in linear regression and logistic regression settings to demonstrate the when GN or Adam may perform better, together with empirical simulations.

**Strengths:**

1. The paper studies an interesting topic, considering the effectiveness of recent preconditioning optimizers like Shampoo and SOAP. It will be nice to find out which way of doing preconditioning is the best, i.e., using GN or Adam, using $ p = 1/2 $ or $ 1 $.
2. The paper starts from simple but important settings, i.e., linear regression and logistic regression, aiming to provide understanding through theoretical analysis in these relatively simple settings.

**Weaknesses:**

1. My major concern is that the results are basically not strong enough. For the major contributions and claims, it seems not clear what conclusions we can draw from the paper. Could the authors provide some clear conclusions from the paper on maybe when we should use Adam and when we should use GN, or maybe when we should use $ p=1/2 $ and when we should use $ p = 1 $? To me, the theoretical part of the paper majorly provides some extreme cases as examples for that Adam may outperform GN and GN may out perform Adam, without a proper explanation and analysis for how these examples can be reflected in the real applications of Adam and GN.
2. The experiments don't seem to be valid for me. Simulations for the examples are good, but 2-layer MLP and 1-layer transformer are definitely not enough for validation of real nonconvex settings.
3. I suggest the authors better introduce and organize the notations in the paper, which may not be easy to follow currently. For example, the notation $ g(x) $ and even $ x $ itself lacks a proper definition, but appears very often and plays an important role.

**Questions:**

1. I am curious about why comparing GN with Adam? And what can we learn from this comparison?

---

> ### Author Response · Authors · 2025-11-18
>
> We thank the reviewer for the helpful feedback! We will start with reiterating the main motivation and contribution of the paper, which may help resolve potential misunderstandings.
>
> - **Why comparing Adam and Gauss-Newton (GN)**: Adam is the de-facto optimizer used in practice, which enjoys great empirical success despite having insufficient theoretical backing. On the other hand, Gauss-Newton methods tie more closely to the theoretically justified Newton’s method, though can be computationally infeasible in practice. Therefore, toward a better-informed optimizer design, it is of great importance to understand whether Adam is merely a computationally cheaper comprise, or can it outperform GN in some cases — our results confirm the latter, as detailed below.
> - **Our main contributions:** by separating the consideration of the diagonal preconditioner choice and the basis choice, we show that the comparison is subtly affected by basis and gradient noise, as summarized by the theoretical results in Table 1. These results suggest that the comparison for Adam vs Gauss-Newton is subtle, which is why there may not be a simple prescription for which algorithm would be better in an arbitrary setting.
> Nevertheless, based on our current empirical results, we hypothesize that MLP experiments generally agree with the linear regression results, and Transformer experiments, due to the softmax function in attention, likely align better with logistic regression where Adam can outperform Gauss-Newton even under the identity basis. The actual comparison for a particular practical setup will depend on the exact data and architecture choice as the reviewer also alluded to, and we agree that it would be an interesting future direction.
>
> Regarding **connection to practice**, as discussed above, our analyses on the quadratic model and logistic regression complement each other in providing a more complete picture. Moreover, the auto-tuning effect of Adam in Sec 3.1.2 is a concrete takeaway that we believe to be generically applicable. The auto-tuning effect, combined with Corollary 1 showing that such an effect does not exist at small batch sizes, suggests that using the full-batch gradient as preconditioner could be useful at small batch sizes, which is consistent with a concurrent work [1].
>
> Thank you also for suggesting **notation improvements**. Following the convention, we use $g$ to denote the gradient (defined on line 106) and use $x$ to denote the input to the function (defined on line 149). We have modified the revision (line 101, 150; highlighted in blue) to make this clear. If there’re any suggestions to help improve the notational clarity, please let us know.
>
> Please let us know if you have other comments or suggestions, and thanks again for your review!
>
> [1] Roulet and Agarwala 2025: Per-example gradients: a new frontier for understanding and improving optimizers.

---

### Author Response · Authors · 2025-12-02
**Rebuttal summary**

We thank the reviewers for their helpful feedback and would like to summarize the discussion below.

Our work compares two common adaptive methods, namely Adam vs Gauss-Newton (GN), which all reviewers find to be a critical and relevant problem.

- We propose to disentangle the preconditioner’s design choices into 1) the **basis choice** (e.g. the eigenbasis of the GN matrix vs the canonical basis), and 2) the **diagonal preconditioner choice** within the basis, i.e. Adam vs GN (eq 2,3 in the paper).
    - This framework is generally applicable across optimizers, which the reviewers consider as an important contribution and a “novel and powerful lens” (Reviewer UWDe).
    - We clarified (in response to Reviewer UWDe) that in our paper, the GN preconditioner refers to a specific choice of the *diagonal* preconditioner (independent of the basis choice), as opposed to a conventional use of the term which takes the GN matrix as the preconditioner (setting the basis choice to be the eigenbasis of the GN matrix).
- Under this framework, we show that the comparison of diagonal preconditioners (i.e. Adam vs (diagonal) GN) is subtly affected by both the basis choice and the **gradient stochasticity** (i.e. full batches vs small batches).
    - Our **theoretical analyses** (Table 1) are based on linear and logistic regression, which are classic setups that are “simple but important” (Reviewer VcjS, mAGy).
    - Our experiments confirm that our theoretical results have broader **practical implications**, which some reviewers (Reviewer VcjS, UWDe, sdk6) asked about this. To summarize:
        - Our MLP experiments are consistent with the conclusions from linear regression. In particular, Adam’s **auto-tuning effect** (Sec 3.1.2) is a concrete takeaway that we believe to be generally applicable and “worth highlighting” (Reviewer mAGy).
        - Our Transformer experiments are consistent with the conclusion from logistic regression, where Adam can “surprisingly” (Reviewer mAGy) outperform GN even under the eigenbasis with full-batch updates. The connection between Transformer and logistic regression is likely due to the softmax function in attention.

The remaining comments are related to clarifications, which we believe have been addressed in our rebuttal responses.

---

### Meta-Review · Area_Chair_upfA · 2025-12-26

**Summary:**

This paper investigates the differences between the Adam preconditioner and the Gauss-Newton preconditioner. The paper focuses its analysis on two axes: choice of basis and gradient noise (i.e. the effect of the batch size). The reviewers agree that the topic is relevant, but differ significantly in how convincing they find the conclusions and how far the results generalize. One reviewer is strongly positive (clear accept) while all the other reviewers find the ideas interesting but the conclusions insufficiently validated. One predominant concern raised by the reviewers is that the conclusions rely on simple models while the experiments are conducted on rather shallow networks, so it seems difficult to judge whether the practical implications for modern architectures and realistic basis misalignment scenarios hold in practice.

As a minor comment, the section on choice of basis would benefit from providing stronger justification for several claims, e.g.
"it is well-known that GN-1 achieves the optimal convergence rate in both full-batch and stochastic setting"
The appendix only provides a proof for quadratic functions (note the small typo there: $\theta$ should be $\theta^{(0)}$).
Relevant works include the following thesis: https://publikationen.uni-tuebingen.de/xmlui/handle/10900/128019, as well as the work of Adolph et al. cited in the thesis.

The authors did answer many of the technical questions in the rebuttal. However, they did not add new experiments that would have directly addressed one of the main recurrent criticisms raised by the reviewers. As a result, it is unlikely that the reviewers' scores would substantially increase, and I therefore cannot recommend acceptance at this time. That said, I find the research direction promising and encourage the authors to further strengthen the manuscript, and I would be interested in seeing a revised version in the future.

**Reviewer Concerns:**

My main concern is that the authors did not attempt to address one of the main criticisms, namely the insufficient validation on non-toy neural network architectures. The authors simply say that they "conjecture" that the conclusion should extend. While I do not like to ask for lots of additional and unnecessary experiments, it feels that the bar set in this paper is much lower than what one would expect nowadays.

**Reviewer Scores:**

I addressed this in my general response. One of the main criticisms remains so I doubt they would have increased their scores.

---

### Decision · Program_Chairs · 2026-01-26

Reject